# SageAttention2: Efficient Attention with Thorough Outlier Smoothing and Per-thread INT4 Quantization

Jintao Zhang [*1]  Haofeng Huang [*12]  Pengle Zhang [1]  Jia Wei [1]  Jun Zhu [1]  Jianfei Chen [1]

https://github.com/thu-ml/SageAttention

## Abstract

Although quantization for linear layers has been widely used, its application to accelerate the attention process remains limited. To further enhance the efficiency of attention computation compared to SageAttention while maintaining precision, we propose `SageAttention2`, which utilizes significantly faster 4-bit matrix multiplication (Matmul) alongside additional precision-enhancing techniques. First, we propose to quantize matrices $(Q, K)$ to INT4 in a hardware-friendly thread-level granularity and quantize matrices $(\widetilde{P}, V)$ to FP8. Second, we propose a method to smooth $Q$, enhancing the accuracy of INT4 $QK^\top$. Third, we propose a two-level accumulation strategy for $\widetilde{P}V$ to enhance the accuracy of FP8 $\widetilde{P}V$. The operations per second (OPS) of `SageAttention2` surpass FlashAttention2 and xformers by about **3x** and **4.5x**. Moreover, `SageAttention2` matches the speed of FlashAttention3(fp8) on the Hopper GPUs, while delivering much higher accuracy. Comprehensive experiments confirm that our approach incurs negligible end-to-end metrics loss across diverse models, including those for language, image, and video generation. The code is available at https://github.com/thu-ml/SageAttention.

## 1. Introduction

Due to the quadratic time complexity of attention, its efficient implementation becomes crucial as sequence lengths

---
[*]Equal contribution  [1]Dept. of Comp. Sci. and Tech., Institute for AI, BNRist Center, THBI Lab, Tsinghua-Bosch Joint ML Center, Tsinghua University [2]Institute for Interdisciplinary Information Sciences, Tsinghua University. Correspondence to: Jianfei Chen <jianfeic@tsinghua.edu.cn>.

*Proceedings of the 42nd International Conference on Machine Learning*, Vancouver, Canada. PMLR 267, 2025. Copyright 2025 by the author(s).

increase in real-world applications (Jiang et al., 2024; Zhang et al., 2025b). Several strategies have been developed to mitigate the computational demands of attention —such as (1) *Linear attention* methods (Wang et al., 2020; Choromanski et al., 2021; Yu et al., 2022; Katharopoulos et al., 2020) that reduce complexity to $O(N)$ and (2) *Sparse attention* methods (Liu et al., 2021; Chu et al., 2021; Li et al., 2022; Xiao et al., 2024b;a; Chen et al., 2024; Jiang et al., 2024; Venkataramanan et al., 2024; Gao et al., 2024; Fu et al., 2024; Zhang et al., 2025e; Xi et al., 2025; Yang et al., 2025a; Zhang et al., 2025f;h;i) that selectively process parts of the context — these methods are only suitable for a limited range of models and tasks. The widely used attention methods optimize attention by exploiting hardware capacities, such as FlashAttention V1, V2, V3 (Dao et al., 2022; Dao, 2024; Shah et al., 2024), xformers (Lefaudeux et al., 2022), and SageAttention (Zhang et al., 2025c;d;g;a). These works do not omit computations across the entire sequence and achieve impressive speed and accuracy performance across various tasks.

**Motivation**. For the two matrix multiplication (Matmul) operations in attention: $QK^\top$ and $\widetilde{P}V$, SageAttention accelerates them by quantizing the $QK^\top$ to INT8 and uses FP16 Matmul with FP16 accumulators for $\widetilde{P}V$. Moreover, to maintain attention accuracy, SageAttention proposes smoothing $K$ by eliminating its channel-wise outliers. SageAttention achieves $2 \times$ speedup compared with FlashAttention2 and is the first quantized attention that incurs a negligible end-to-end metrics loss across language, image, and video generation models. However, SageAttention has two weaknesses. **(W1)** INT8 Matmul achieves only half the speed of INT4. **(W2)** FP16 Matmul with FP16 accumulators provides a speedup only on RTX 4090 and RTX 3090 GPUs. To leverage the faster INT4 tensor cores for $QK^\top$ and using a method that can accelerate $\widetilde{P}V$ on a broader range of GPUs, we propose to quantize $Q, K$ to INT4 and $\widetilde{P}, V$ to FP8.

**Challenges**. Quantizing $Q, K$ to INT4 and $\widetilde{P}, V$ to FP8 presents significant challenges. For example, when only per-tensor quantizing $Q, K$ to INT4, the text-to-video model CogvideoX will generate a completely blurry video, and

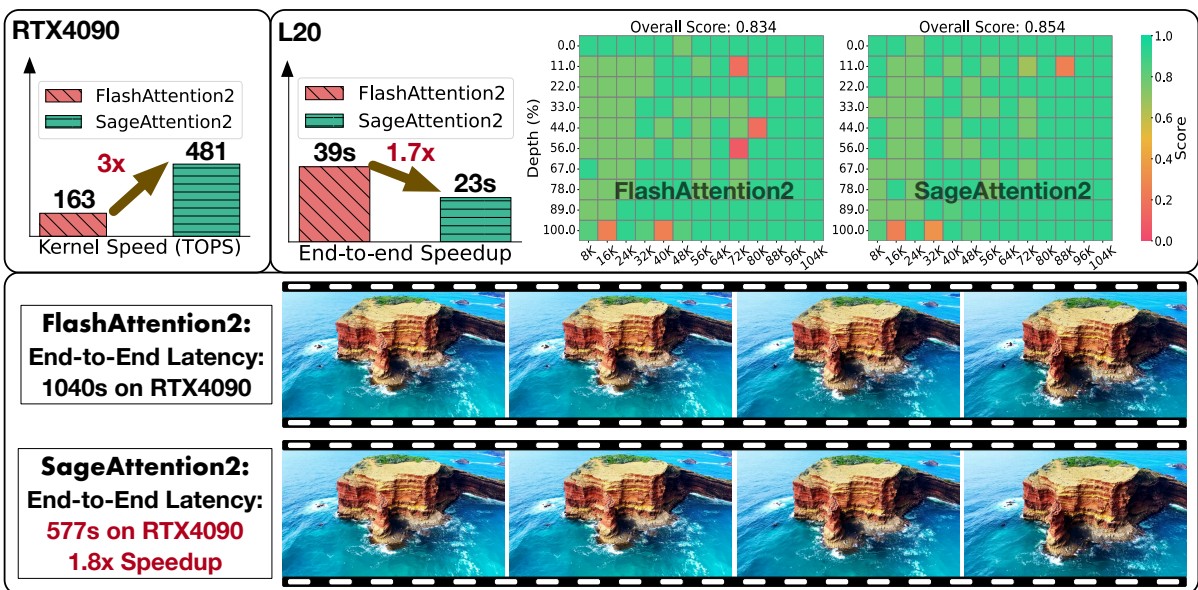

Figure 1. The upper left figure shows the kernel speedup on RTX4090 GPU. The upper right figure shows the end-to-end inference speedup of generating the first token and performance metrics for the Needle-in-a-Haystack task (Kamradt, 2023) with a sequence length of 100K on `Llama3.1` on L20 GPU. The figure below shows two videos generated by `CogvideoX` (1.5-5B) using FlashAttention2 and `SageAttention2` on RTX4090, showing that `SageAttention2` accelerates generation by 1.8x with no video quality loss.

Llama3 only achieves a random-guessing-level accuracy of 25% on MMLU. After an in-depth investigation, we identified three primary challenges: **(C1)** The numerical range of INT4 is quite limited compared to FP16 or INT8, leading to significant quantization errors when $Q$ and $K$ have some abnormal values. **(C2)** We discover that the FP32 accumulator designed for FP8 matrix multiplication in the tensor core (`mma.f32.f8.f8.f32`) is actually FP22, specifically with 1 sign bit, 8 exponent bits, and 13 mantissa bits. This will lead to an accuracy loss of $\widetilde{P}V$.

**Our approach**. To address **(C1)**, we first discover that the per-block quantization of $Q, K$ in SageAttention does not achieve sufficient accuracy for INT4 quantization. Simultaneously, to avoid the extra latency caused by per-token dequantization, where each GPU thread corresponds to multiple quantization scales, we propose an adequately precise quantization method that incurs no additional dequantization overhead. Specifically, we introduce a per-thread quantization approach based on the mapping between the GPU threads and the memory layout of matrices as dictated by the PTX `mma` instructions. This method groups tokens corresponding to the same thread for quantization and dequantization, ensuring that each thread is associated with a single quantization scale. This approach achieves much better accuracy performance than per-block quantization with no additional dequantization overhead. Second, for the significant channel-wise outliers in matrices $Q$ and $K$, we adopt smoothing $K$ in SageAttention and further propose an effective method to remove these outliers in $Q$. Specifically, we propose subtracting the average value of the channel dimen-

sion of $Q$, referred to as $\overrightarrow{Q}_m$. Then, we add $\overrightarrow{Q}_m K$ after the $QK^\top$ Matmul to ensure the correctness of the attention computation. To address **(C2)**, the accuracy loss caused by the 22-bit accumulator for FP8 Matmul of $\widetilde{P}V$, we propose a two-level accumulation strategy that uses an FP32 buffer to accumulate the values from the 22-bit accumulator after each block Matmul of $\widetilde{P}V$. This method confines the errors to the block range. Additionally, we design an optional technique to enhance the accuracy of the 22-bit accumulator. Specifically, we could smooth $V$ by subtracting the average value of its channel dimension and adding the subtracted item to the attention output to maintain the correctness.

**Performance.** We offer a high-performance implementation of `SageAttention2` on RTX4090 and L20 GPUs. This implementation achieves a peak performance of **481 TOPS** on the RTX4090, outperforming FlashAttention2 and xformers by approximately 3x and 4.5x, respectively. To support NVIDIA Hopper GPUs, which lack native INT4 tensor core support, we additionally provide `SageAttention2-8b`, a variant that quantizes $Q, K$ to INT8. FlashAttention3, in contrast, is tailored to and only compatible with the Hopper architecture. Moreover, `SageAttention2-8b` matches the speed of FlashAttention3(fp8) on Hopper GPUs, while delivering much better accuracy. For example, on popular video generation models, our method does not compromise end-to-end accuracy, whereas FlashAttention3(fp8) brings noticeable degradation, as visualized in Fig. 7 and 9. We extensively evaluate the end-to-end metrics of state-of-the-art text, image, and video generation models. `SageAttention2` can accelerate models in a plug-and-

play way with negligible loss in end-to-end metrics.

## 2. Preliminary

### 2.1. FlashAttention

The attention computation can be formulated as: $S = QK^\top/\sqrt{d}$, $P = \sigma(S)$, $O = PV$, where $\sigma(S)_{ij} = \exp(S_{ij})/\sum_k \exp(S_{ik})$. The matrices $Q$, $K$, and $V$ each have dimensionality $N \times d$, and $S$, $P$ are $N \times N$. $d$ is typically small, e.g., 64 or 128, and $N$ can be thousands or millions. The time complexity of attention is $O(N^2)$, primarily due to two matrix multiplications ($QK^\top$ and $PV$), both with complexities of $O(N^2d)$. FlashAttention (Dao, 2024) is a GPU-friendly attention implementation, which tiles $Q$, $K$, and $V$ from the token dimension into blocks $\{Q_i\}_{i=1}^{n_q}, \{K_i\}_{i=q}^{n_k}, \{V_i\}_{i=1}^{n_v}$ with block sizes of $b_q, b_k, b_v$ tokens, respectively, where $n_q, n_k, n_v$ are the number of tiles, and $b_k = b_v$. FlashAttention computes the output matrix $O$ in parallel in tiles. Each streaming multiprocessor (SM) computes a block $O_i$ (corresponds to a $Q_i$) by iteratively loads $K_j, V_j$ for each $j$, and update the output with online softmax (Milakov & Gimelshein, 2018):

$$S_{ij} = Q_i K_j^\top/\sqrt{d}, \quad (m_{ij}, \widetilde{P}_{ij}) = \tilde{\sigma}(m_{i,j-1}, S_{ij}), \quad (1)$$
$$l_{ij} = \exp(m_{i,j-1} - m_{ij})l_{i,j-1} + \text{rowsum}(\widetilde{P}_{ij}),$$
$$O_{ij} = \text{diag}\left(\exp(m_{i,j-1} - m_{ij})\right)O_{i,j-1} + \widetilde{P}_{ij}V_j,$$

where $m_{ij}$ and $l_{ij}$ are $b_q$-dimenalional vectors, initialized with $-\infty$ and 0 respectively. $\tilde{\sigma}(\cdot)$ is an online softmax operator: $m_{ij} = \max\{m_{i,j-1}, \text{rowmax}(S_{ij})\}$, $\widetilde{P}_{ij} = \exp(S_{ij} - m_{ij})$. Finally, the output is computed as $O_i = \text{diag}(l_{i,n_q})^{-1}O_{i,n_q}$.

### 2.2. Quantization

A matrix multiplication $C = AB$ can be accelerated with quantization as:

$$(\delta_A, \hat{A}) = \psi(A), \quad (\delta_B, \hat{B}) = \psi(B), \quad C = \psi_{\delta_A \delta_B}^{-1}(\hat{A}\hat{B})$$

$\psi$ is a *quantizer* which converts a high-precision (e.g., FP32 or FP16) matrix $A$ to a low-precision format $\hat{A}$ (e.g., INT4 or FP8) with a *scale* $\delta_A$, and $\psi^{-1}$ is a *dequantizer* to convert back to high-precision. We should have $\psi_{\delta_A}^{-1}(\hat{A}) \approx A$. The actual matrix multiplication $\hat{A}\hat{B}$ is performed with low precision. In modern GPUs, low-precision matrix multiplication is usually multiple times faster than higher-precision ones. Quantizers differ in *numerical format* and *granularity*, e.g., how many elements ("quantization group") share a common scale factor. For example, an *INT4, per-tensor quantizer* first computes the scale as the maximum absolute value of the entire tensor, scales the elements to the maximum representable range of INT4 [-7, +7], and then casts to INT4 with rounding:

$\hat{A} = \lceil A/\delta_A \rfloor, \delta_A = \max(|A|)/7$. The dequantization process is an element-wise scaling. For example, for per-tensor dequantization, $\psi_{\delta_A \delta_B}^{-1}(\hat{A}\hat{B}) = \hat{A}\hat{B} \times \delta_A \delta_B$.

*Table 1.* Speedup compared to matrix multiplication in FP16 with an FP32 accumulator.

| GPU | MM Input | MM Accumulator | Speedup |
|---|---|---|---|
| RTX4090 | FP16 | FP16 | 2x |
| | FP8 | FP32 | 2x |
| L40, L20 | FP16 | FP16 | **1x** |
| H100 | FP8 | FP32 | 2x |

### 2.3. SageAttention

Based on the block tiling in FlashAttention (Dao et al., 2022), SageAttention (Zhang et al., 2025c) quantizes $Q, K$ to INT8 in a per-block granularity, that is, each $Q_i, K_i$ has a separate scalar scale: $\delta_{Q_i} = \max(|Q_i|)/127, \delta_{K_j} = \max(|K_j|)/127$. In this way, the product $S_{ij}$ in Eq. (1) can be approximated as $S_{ij} \approx \hat{Q}_i \hat{K}_j^\top \times (\delta_{Q_i}\delta_{K_j}/\sqrt{d})$. To maintain accuracy, SageAttention proposes a preprocessing technique by subtracting the token-wise mean from $K$. Additionally, SageAttention keeps $\widetilde{P}_{ij}$ and $V_j$ in FP16, but utilizes an FP16 accumulator (rather than FP32) for computing the product $\widetilde{P}_{ij}V_j$. Reducing accumulator precision can accelerate matrix multiplication (MM) on the RTX4090 GPU. However, other GPUs, such as L20, L40, or H100, do not exhibit this behavior, as shown in Table 1.

## 3. SageAttention2

In this section, we introduce SageAttention2, an efficient and accurate quantized attention. The workflow of SageAttention2 is shown in Fig. 3. We quantize $Q, K$ to INT4 and $\tilde{P}, V$ to FP8 to maximize the efficiency and propose several techniques, including $QK$-smoothing, per-thread quantization, and two-level accumulation to preserve the accuracy, which we shall discuss in subsequent subsections.

### 3.1. Smooth $Q$

First, we discuss how to accurately compute $QK^\top$ with INT4. The numerical range of INT4 is notably restrictive. This affects quantization due to the presence of *outliers* (Lin et al., 2025). Given the INT4 range [-7, +7], any element will be quantized to zero if it is more than 14 times (0.5 vs 7) smaller than the largest element in the group. Since outliers are much larger than other elements, it is likely that many non-outlier elements are quantized to zero, leading to substantial accuracy degradation. Therefore, to keep the quantization accurate, we need to keep the largest element small, making the magnitude of elements as uniform as possible. Such technique is called *smoothing*.

Here, we propose a smoothing technique inspired by

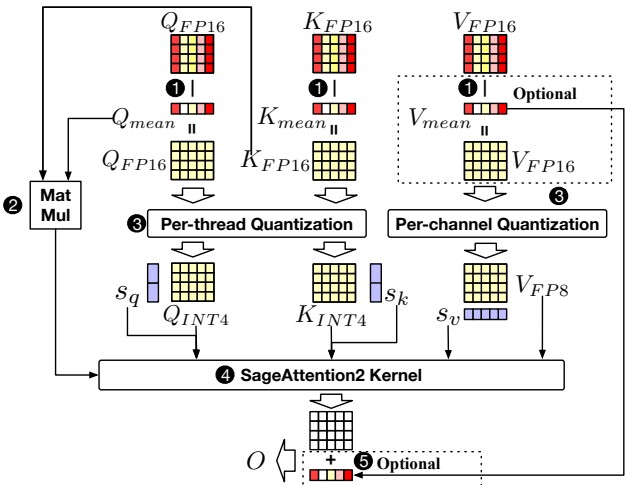

*Figure 2.* Typical examples of tensors' data distribution in attention.

*Figure 3.* Workflow of `SageAttention2`. ❶ Smooth $Q, K, V$. ❷ A GEMV to obtain $\Delta S$. ❸ Per-thread quantize $Q, K$ and per-channel quantize $V$. ❹ Perform the `SageAttention2` kernel. ❺ Correct the output.

SageAttention (Zhang et al., 2025c). SageAttention observed that $Q, K$ for all tokens are actually highly similar, with only small variations between different tokens (Fig. 2 shows the heatmap distributions of the $Q$, $K$, and $V$ randomly sampled from `Llama3.1` (Dubey et al., 2024) and `CogvideoX` (Yang et al., 2025b)). We propose to smooth $K$ as SageAttention does and further smooth $Q$ by subtracting a common mean of each block:

$$\gamma(Q_i) = Q_i - \bar{q}_i, \quad \gamma(K_j) = K_j - \bar{k}, \quad (2)$$

where $\bar{q}_i = \mathrm{mean}(Q_i), \bar{k} = \mathrm{mean}(K)$ are $1 \times D$ vectors, the mean is conducted along the token axis, and $\bar{q}_i, \bar{k}$ are broadcasted to tokens in a block and a tensor for subtraction.

With the decomposition, we have $S_{ij} = Q_i K_j^\top = (\bar{q}_i + \gamma(Q_i))(\bar{k} + \gamma(K_j))^\top = \bar{q}_i \bar{k}^\top + \bar{q}_i \gamma(K_j)^\top + \gamma(Q_i) \bar{k}^\top + \gamma(Q_i) \gamma(K_j)^\top = \gamma(Q_i) \gamma(K_j)^\top + \Delta S_{ij} + b$. Here, $\Delta S_{ij} = \bar{q}_i \gamma(K_j)^\top$ is an $1 \times N$ vector, and $b = \bar{q}_i \bar{k}^\top + \gamma(Q_i) \bar{k}^\top$ is an $N \times 1$ vector. We do not need to compute $b$ since adding a common bias to an entire row of $S$ does not affect the result after softmax. Therefore, we can accelerate $Q_i K_j^\top$ with INT4 by the following two stages:

(1) *preprocessing*: smooth $Q, K$ according to Eq. (2), apply quantization $(\delta_{Q_i}, \hat{Q}_i) = \psi_Q(\gamma(Q_i)), (\delta_{K_j}, \hat{K}_j) = \psi_K(\gamma(K_j))$, and compute $\Delta S_{ij} = \bar{q}_i \gamma(K_j)^\top$. Smoothing,

quantization, and GEMV (general matrix-vector multiplication) for computing $\Delta S$ can be fused into a single kernel, which reads the off-chip $Q$ and $K$ only once.

(2) *attention*: execute the low-precision GEMM, dequantize, and add back the vector $\Delta S$: $S_{ij} = \psi_{\delta_{Q_i} \delta_{K_j}}^{-1}(\hat{Q}_i \hat{K}_j^\top) + \Delta S_{ij}$. These operations are all done on chip, and the dequantization and vector addition only add a marginal overhead compared to the expensive `mma` operation for MM. Importantly, $\gamma(Q_i), \gamma(K_j)$ are quantized rather than $Q_i, K_j$. Since the smoothed matrices are much smaller in magnitude and contain fewer outliers, the quantization accuracy can be significantly improved. A theoretical analysis of the benefit of smoothing is included in Appendix A.5.

**Remark.** Classical techniques to improve the activation-weight MM, such as per-channel quantization, or SmoothQuant (Xiao et al., 2023) are not applicable here for the query-key MM in attention. Per-channel quantization cannot be applied to $Q, K$ because the quantization must be conducted along the outer axis (token dimension) of $QK^\top$. On the other hand, both $Q$ and $K$ have significant outliers, so trading the quantization accuracy between them with SmoothQuant cannot work effectively, as shown in Sec. 4. Here, we utilize the unique token similarity pattern in attention to derive a dedicated quantization method for $Q$ and $K$. The previous work SageAttention only smooths $K$, so it is less accurate than our method.

**Empirical results.** Fig. 20 in Appendix A.9 shows an example from `CogvideoX` of the distribution of $\hat{Q}$ with and without smoothing $Q$. We can find that with smoothing $Q$, the range of INT4 is utilized more uniformly and fully. Table 5 presents end-to-end metrics for different quantization methods with and without *smoothing Q+K* on `Llama3.1` and `CogvideoX` (2b). The results demonstrate that *smoothing Q+K* offers significant accuracy benefits. Also, Table 4 and 17 show that the order of effectiveness is *smoothing Q+K > smoothing Q > smoothing K > other baselines*.

### 3.2. INT4 Per-thread Quantization

Orthogonal to smoothing, we can mitigate the problem of outliers by refining the quantization granularity so that the number of affected elements by outliers becomes smaller. Although per-token quantization offers a detailed level of granularity, it results in significant overhead

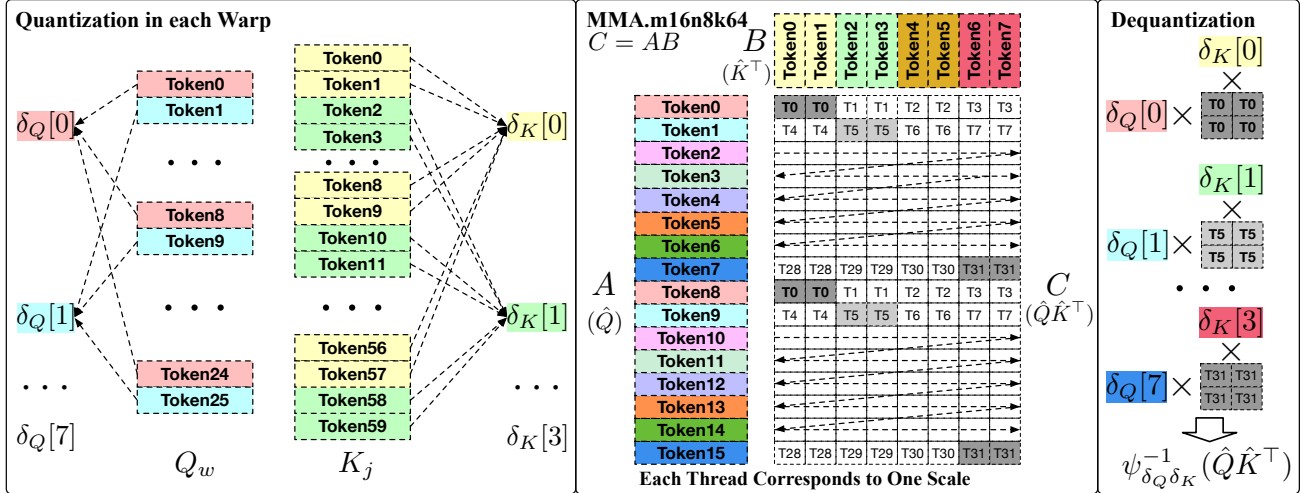

*Figure 4.* An example of per-thread quantization. The left figure shows the correspondence between the quantization scales and the tokens in each GPU warp. The right figure shows the correspondence between quantization tokens and GPU threads in a `MMA.m16n8k64` instruction, showing that each GPU thread only corresponds to one quantization scale in $\delta_Q$ and $\delta_K$ in dequantization.

during dequantization. Specifically, each GPU thread in per-token quantization must handle multiple quantization scales, leading to a high latency of the dot product of the quantization scale vectors $\delta_Q$ and $\delta_K$. SageAttention uses per-block quantization, where each block $Q_i$ ($b_q$ tokens) and $K_i$ ($b_k$ tokens) have a single quantization scale. Such a quantization strategy could achieve an accuracy performance close to per-token quantization and avoid the high dequantization overhead. However, quantizing $Q$ and $K$ to INT4 demands a finer quantization granularity. To address this, we propose *per-thread quantization*, a more precise and granular approach than the *per-block quantizer*, also without the additional overhead of the vector dot product between $\delta_Q$ and $\delta_K$.

Specifically, each block of $Q$, i.e., $Q_i$, in SageAttention will be split into $c_w$ segments and processed by $c_w$ GPU warps in a GPU streaming processor (SM). We call each segment of $Q_i$ as $Q_w$, and $k_w = K_j$ since $K_j$ is shared among warps. Then, each warp containing 32 threads uses the `mma.m16n8k64` PTX instruction (NVIDIA) for the $Q_w K_j^\top$. According to the layout requirement of this instruction, we find that $Q_w[8k + i]$ could share one quantization scale, and $K_j[8k + 2i]$ together with $K_j[8k + 2i + 1]$ could share one quantization scale. Such a quantization method is more fine-grained with no additional overhead. This is because it assigns different GPU threads to distinct quantization groups based on the MMA instruction layout, with each thread performing dequantization only using a single quantization scale value. We show an example of per-thread quantization in Fig. 4. The detailed formulation is shown in Equation 8 and Fig. 18 (please see Appendix A.6 for more detail).

**Empirical results.** As shown in Table 6 and Table 15, we compare the average and worst accuracy of INT4 quantization at per-token, per-thread, per-block, and per-tensor granularity using real $Q, K, V$ across all layers of `CogvideoX`. Results indicate that the accuracy of per-thread quantization is very close to per-token and significantly outperforms other granularities. Moreover, Table 19 shows that per-thread quantization incurs almost no speed degradation, while per-token quantization introduces noticeable overhead due to the reduced hardware efficiency.

### 3.3. FP8 quantization for $\tilde{P}V$

We now turn to the MM $\tilde{P}V$, where $\tilde{P}_{ij} = \exp(S_{ij} - m_{ij})$ is the unnormalized quantity according to Eq. (1). The distribution of $\tilde{P}$ is unique and differs from other activations. First, we note that $S_{ij} - m_{ij} \leq 0$, so $P_{ij} \in [0, 1]$ ($\leq$ and $\in$ apply element-wise). We find that $\tilde{P}$ often consists of many small elements, but their sum is non-negligible (e.g., 5000 elements around $10^{-4}$). In this case, we must represent small elements accurately. INT quantization is unsuitable for this setting, since it distributes the quantization points evenly within the numerical range. SageAttention (Zhang et al., 2025c) choose to retain $\tilde{P}$ and $V$ in FP16, and accelerate the MM by decreasing the accumulator precision. However, this strategy is only effective on very few GPUs.

We propose to quantize $\widetilde{P}, V$ to FP8 with 4 exponent bits and 3 mantissa bits (E4M3). The numerical range of E4M3 is $[-448, +448]$. We quantize $P$ with a static scale: $\delta_P = \frac{1}{448}$ since the original $P$ elements are already in $[0, 1]$. We quantize $V$ per-channel to address the channel-wise outliers shown in Fig. 2. Empirical results in Table 7 and Table 16 show the average and worst accuracy of different

---

**Algorithm 1** Implementation of SageAttention2.

---

**Input:** Matrices $Q(\text{FP16}), K(\text{FP16}), V(\text{FP16}) \in \mathbb{R}^{N \times d}$, block size $b_q, b_{kv}$, warp count $c_w$.
**Preprocessing:** $K = K - \text{mean}(K)$, $(\delta_V, \hat{V}) = \psi_V(V)$. //per-channel.
Divide $Q$ to $T_m = N/b_q$ blocks $\{Q_i\}$; divide $K$, and $V$ to $T_n = N/b_{kv}$ blocks $\{K_i\}, \{V_i\}$;
**for** $i = 1$ **to** $T_m$ **do**
  $\bar{q}_i = \text{mean}(Q_i)$, $(\delta_Q, \hat{Q}_i) = \psi_Q(Q_i - \bar{q}_i)$ //per-thread ;
  **for j** in $[1, T_n]$ **do**
    $(\delta_K, \hat{K}_j) = \psi_K(K_j)$ //per-thread, $\quad w = \text{range}(c_w), st = w * c_w$ ;
    $S_{ij}[st : st + c_w] = \psi_{\delta_Q \delta_K}^{-1}(\text{Matmul}(\hat{Q}_i[st : st + c_w], \hat{K}_j^\top)) + \text{GEMV}(\bar{q}_i, K_j^\top)$ ; // Paralleled by $c_w$ warps. The $\psi_{\delta_Q \delta_K}^{-1}$ is illustrated in Fig. 4.
    $m_{ij} = \max(m_{i,j-1}, \text{rowmax}(S_{ij}))$, $\widetilde{P}_{ij} = \exp(S_{ij} - m_{ij})$, $l_{ij} = e^{m_{i,j-1} - m_{ij}} + \text{rowsum}(\widetilde{P}_{ij})$ ;
    $O_{ij}(\text{FP22}) = \text{Matmul}((\widetilde{P}_{ij} * 448).\text{to}(\text{FP8.e4m3}), V_j)$ ;
    $O_{ij}(\text{FP32}) = \text{diag}(e^{m_{i,j-1} - m_{ij}})^{-1} O_{i,j-1}(FP32) + O_{ij}(\text{FP22})$ ;
  **end for**
  Load $\delta_V$ into an SM ; $\quad O_i = \text{diag}(l_{i,T_n})^{-1} O_{i,T_n}(\text{FP32}) /448 * \delta_V$ ; $\quad$ Write $O_i$ ;
**end for**
**return** $O = \{O_i\}$

---

data types used for $\widetilde{P}, V$ across all layers of CogvideoX. The accumulator is always 32-bit. We can see that the accuracy of E4M3 is very close to that of FP16 and superior to E5M2 and INT8. Most modern GPUs have tensor cores that support FP8 Matmul operations, which are twice as fast as those using FP16.

### 3.4. FP32 MMA Buffer for FP22 Accumulator

While FP8 quantization for $\tilde{P}V$ above is theoretically accurate in simulation, we observe that the actual CUDA implementation suffers a consistent accuracy degradation. After narrowing down the problem, we find that the accumulator for the `mma(f32f8f8f32)` instruction on the Ada and Hopper architecture is actually FP22, specifically with 1 sign bit, 8 exponent bits, and 13 mantissa bits. Specifically, for `mma(f32f8f8f32)` instruction $C = AB + D$, where $A, B$ are FP8 matrices and $C, D$ are FP32 matrices, we initialize the $A, B$ to zero and vary $D$ to test the data type of the accumulator. When $D$ is initialized with 1 sign bit, 8 exponent bits, and 13 mantissa bits, the value of $C$ exactly matches $D$. However, when $D$ is initialized with more than 13 mantissa bits, the value of $C$ is equal to $D$ with its least significant 10 mantissa bits zeroed out (i.e., truncated). Consequently, matrix multiplication of $\widetilde{P}V$, quantized to FP8, incurs a certain degree of accuracy loss compared to using an FP32 accumulator.

To mitigate this accuracy loss, we adopt a two-level accumulation strategy, which uses an FP32 buffer to accumulate the values of $\tilde{P}_{ij} V_j$ in FP22. Specifically, we rewrite Eq. (1) as $R_{ij} = \widetilde{P}_{ij} V_j, O_{ij} = \text{diag}\left(\exp(m_{i,j-1} - m_{ij})\right) O_{i,j-1} + R_{ij}$. Here, two sets of accumulators $R_{ij}$ and $O_{ij}$ are maintained in the register. $R_{ij}$ is computed with the `mma(f32f8f8f32)` instruction, providing 22 effective bits, which is sufficient since we only accumulate over a small number of $b_k$ tokens (e.g., $b_k = 64$). Then, $R_{ij}$ is

accumulated to $O_{ij}$ in the high FP32 precision.

**Remark.** The two-level accumulation strategy is also implemented in CUTLASS (NVIDIA, 2023) and Deep-Gemm (DeepSeek-AI et al., 2024) for computing weight-activation products in linear layers. To the best of our knowledge, we are the first to discover and investigate the effect of the FP22 accumulator and implement the two-level accumulation for *attention*.

**Optional smooth V technique.** We also figure out another way to mitigate the accuracy loss due to the FP22 accumulator when $V$ possesses channel-wise biases: $\overrightarrow{V_m} = \text{mean}(V, \text{axis} = 0), V = V - \overrightarrow{V_m}$. Furthermore, to maintain the correctness of the attention computation, it is only necessary to add $\overrightarrow{V_m}$ to the final calculation of $O$: $O = O + \overrightarrow{V_m}$. This is because the sum of each row of the $\widetilde{P}$ matrix equals 1, so $\widetilde{P} \overrightarrow{V_m} = \overrightarrow{V_m}$.

**Remark.** For details on smoothing V, see Appendix A.3. This technique is optional and not employed in our main experiments, as it provides significant benefits only when $V$ exhibits channel-wise bias, which are absent in some models, such as Llama3.1 (see Fig. 2).

## 4. Experiment

**Main result.** SageAttention2 is faster than FlashAttention2 and xformers by about **3x** and **4.5x**. Moreover, SageAttention2 matches the speed of FlashAttn3(fp8) on the Hopper GPUs and is much more accurate than FlashAttn3(fp8). SageAttention2 maintains end-to-end metrics across language, image, and video generation models.

### 4.1. Setup

**Models.** We validate the effectiveness of SageAttention2 across a diverse set of repre-

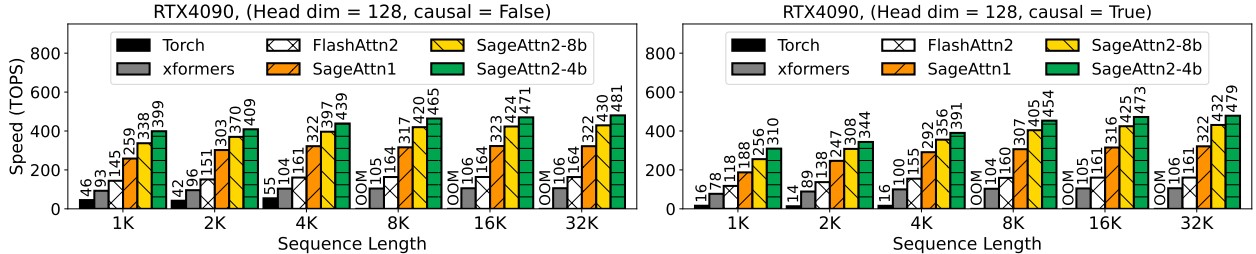

*Figure 5.* Speed comparison between `SageAttention2` and baselines (RTX4090, headdim=128).

*Table 2.* End-to-end metrics across text, image, and video generation models. ✗ indicates an inability to generate results for evaluation.

| Model | Attention | WikiText (Ppl.) ↓ | Lambda (Acc.) ↑ | MMLU (Acc.) ↑ | Longbench ↑ |
|---|---|---|---|---|---|
| Llama3.1 | Full-Precision | 6.013 | 0.815 | 0.635 | 49.40 |
| | HadmdAttn | 7.872 | 0.762 | 0.500 | 44.07 |
| | SmoothAttn | 7.180 | 0.783 | 0.541 | 44.69 |
| | SageAttention | **6.017** | **0.812** | **0.634** | **49.55** |
| | SageAttn2-4b | **6.256** | **0.798** | **0.607** | **48.79** |
| | SageAttn2-8b | **6.019** | **0.811** | **0.634** | **49.59** |
| GLM4 | Full-Precision | 7.241 | 0.432 | 0.743 | 49.78 |
| | HadmdAttn | 7.989 | 0.435 | 0.669 | 45.97 |
| | SmoothAttn | 8.943 | 0.449 | 0.592 | 42.20 |
| | SageAttention | **7.243** | **0.433** | **0.744** | **49.79** |
| | SageAttn2-4b | **7.352** | **0.433** | **0.725** | **49.23** |
| | SageAttn2-8b | **7.242** | **0.432** | **0.745** | **49.60** |

| Model | Attention | CLIPSIM ↑ | CLIP-T ↑ | VQA-a ↑ | VQA-t ↑ | FScore ↑ |
|---|---|---|---|---|---|---|
| CogvideoX (1.5-5B) | Full-Precision | 0.1778 | 0.9979 | 70.231 | 70.928 | 2.507 |
| | HadmdAttn | 0.1576 | 0.9933 | 8.990 | 2.299 | ✗ |
| | SmoothAttn | 0.1559 | 0.9950 | 8.812 | 2.277 | ✗ |
| | SageAttention | ✗ | ✗ | ✗ | ✗ | ✗ |
| | FlashAttn3-fp8 | 0.1562 | 0.9918 | 6.531 | 2.181 | ✗ |
| | SageAttn2-4b | **0.1721** | **0.9978** | **57.729** | **52.989** | **2.884** |
| | SageAttn2-8b | **0.1775** | **0.9980** | **69.492** | **74.415** | **2.487** |
| Hunyuan Video | Full-Precision | 0.1783 | 0.9995 | 82.516 | 75.934 | 0.604 |
| | HadmdAttn | 0.1727 | 0.9989 | 7.514 | 0.762 | 0.175 |
| | SmoothAttn | 0.1739 | 0.9988 | 6.987 | 0.609 | 0.148 |
| | SageAttention | **0.1786** | **0.9995** | **82.496** | **79.843** | **0.597** |
| | FlashAttn3-fp8 | 0.1742 | 0.9941 | 4.433 | 1.460 | ✗ |
| | SageAttn2-4b | **0.1751** | **0.9995** | **81.478** | **65.371** | **0.610** |
| | SageAttn2-8b | **0.1782** | **0.9996** | **81.786** | **75.354** | **0.586** |
| Mochi | Full-Precision | 0.1798 | 0.9986 | 45.549 | 65.416 | 1.266 |
| | HadmdAttn | 0.1733 | 0.9980 | 9.053 | 25.133 | 0.704 |
| | SmoothAttn | 0.1687 | 0.9978 | 3.383 | 3.480 | 0.241 |
| | SageAttention | **0.1800** | **0.9987** | **48.707** | **63.763** | **1.269** |
| | FlashAttn3-fp8 | 0.1762 | 0.9982 | 14.964 | 13.711 | 0.457 |
| | SageAttn2-4b | **0.1783** | **0.9986** | **35.955** | **43.735** | **1.137** |
| | SageAttn2-8b | **0.1797** | **0.9986** | **46.760** | **64.901** | **1.255** |

| Model | Attention | FID ↓ | sFID ↓ | CLIP ↑ | IR ↑ |
|---|---|---|---|---|---|
| Flux | Full-Precision | 10.960 | 16.648 | 26.180 | 1.009 |
| | HadmdAttn | 11.353 | 18.495 | 26.123 | 0.965 |
| | SmoothAttn | 11.149 | 19.017 | 26.109 | 0.959 |
| | SageAttention | **10.944** | **16.641** | **26.171** | **1.008** |
| | SageAttn2-4b | **10.577** | **17.497** | **26.141** | **0.998** |
| | SageAttn2-8b | **10.927** | **16.723** | **26.175** | **1.009** |
| Stable-Diffusion3.5 | Full-Precision | 14.105 | 15.646 | 25.505 | 0.902 |
| | HadmdAttn | 14.259 | 15.909 | 25.513 | 0.886 |
| | SmoothAttn | 14.161 | 15.649 | 25.510 | 0.887 |
| | SageAttention | **14.140** | **15.678** | **25.503** | **0.902** |
| | SageAttn2-4b | **14.097** | **15.397** | **25.487** | **0.895** |
| | SageAttn2-8b | **14.106** | **15.647** | **25.499** | **0.901** |

| Full precision Attention | SageAttention2-8b | SageAttention2-4b | SmoothAttn | HadmdAttn |
|---|---|---|---|---|

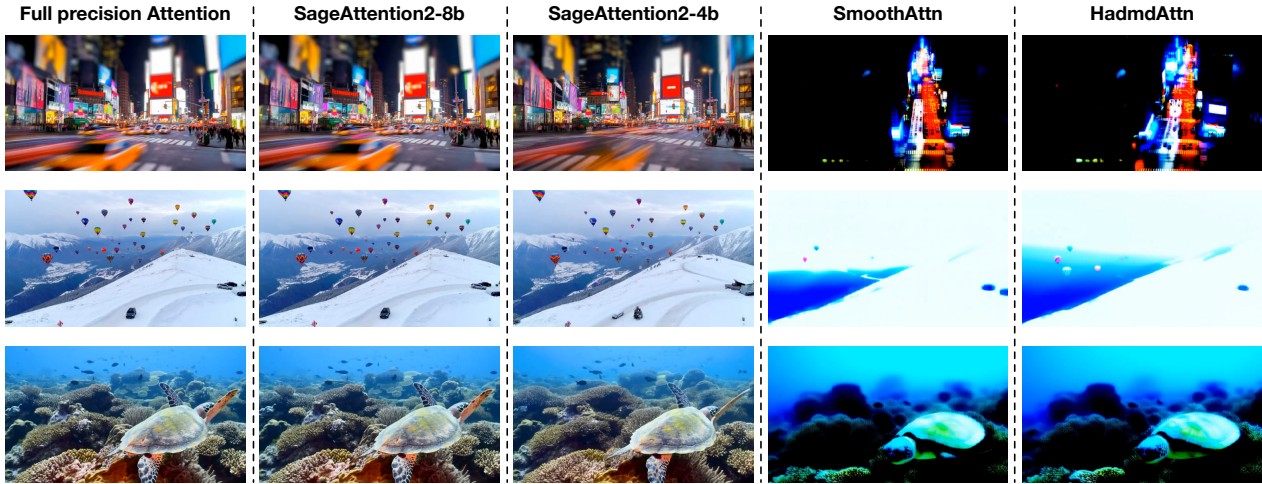

*Figure 6.* Comparison examples from `HunyuanVideo`, prompts are sampled from open-sora prompt sets.

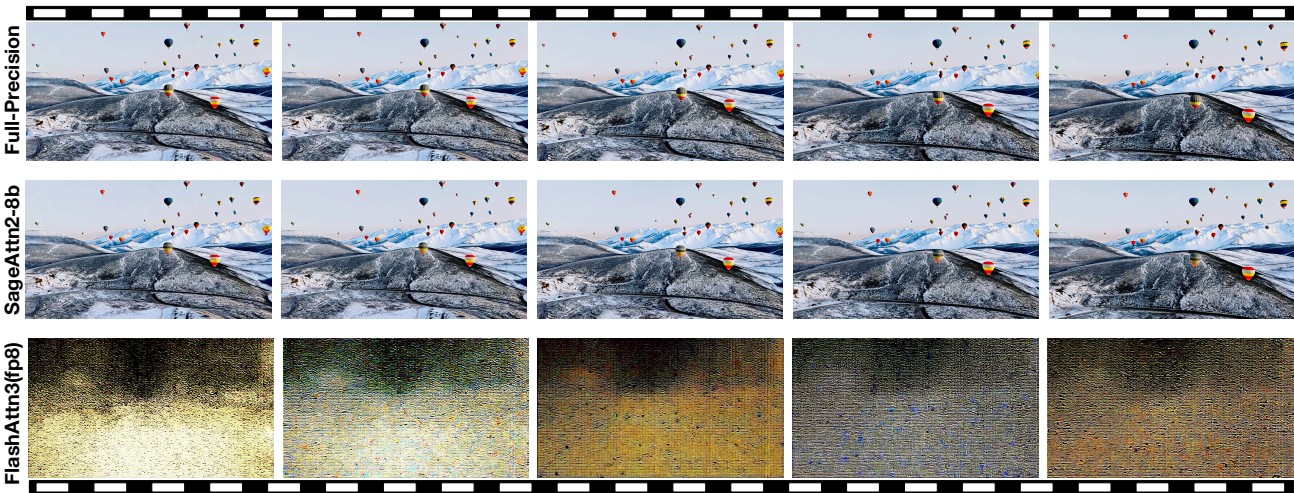

*Figure 7.* A comparison example using SageAttn2-8b and FlashAttention3 on `CogvideoX-1.5`.

sentative models from language, image, and video generation. Specifically, we conduct experiments on ten models: `Llama2` (7B) (Touvron et al., 2023), `Llama3.1` (8B) (Dubey et al., 2024), and `GLM4` (9B) (GLM et al., 2024) for text2text, `CogvideoX` (2B), `CogvideoX` (1.5-5B) (Yang et al., 2025b), `HunyuanVideo` (Kong et al., 2024), and `Mochi` (Team, 2024) for text2video, `Flux` (schnell) (Black Forest Labs, 2023) and `Stable-Diffusion3.5` (turbo) (Stability AI, 2023) for text2image, and `TIMM` (Wightman, 2019) for image classification.

**Datasets and metrics.** For Details about the datasets and metrics we used, please refer to Appendix. A.7.

*Table 3.* Two kernel implementations of `SageAttention2`.

| Kernel | $\psi_Q(Q), \psi_K(K)$ | $\psi_P(\widetilde{P}), \psi_V(V)$ |
|---|---|---|
| SageAttn2-4b | INT4 per-thread | FP8 per-block and per-channel |
| SageAttn2-8b | INT8 per-thread | FP8 per-block and per-channel |

**Implemetation.** We implement two attention kernels as shown in Table 3 using CUDA. The 8-bit variant is adapted

for NVIDIA Hopper GPUs, which lack native INT4 tensor core support, and incorporates all techniques described in Sec. 3 except for smoothing Q.

**Baselines.** (1) `SmoothAttn`. Following Qserve (Lin et al., 2025), we apply smooth quant for $Q, K$ with smoothing factor $\alpha = 0.5$. (2) `HadmdAttn`. Following Quarot (Ashkboos et al., 2024), we apply random Hadamard transformation for $Q, K$ before INT4 quantization. (3) `SageAttention` (Zhang et al., 2025c), which uses smoothing $K$, INT8 per-block quantization for $Q, K$, and FP16 for $\widetilde{P}, V$. (4) `FlashAttn3(fp8)`, the FP8 version of FlashAttention3, which only runs on Hopper GPUs.

### 4.2. Speed and Accuracy of Kernels

**Kernel Speed.** We compare the speed of `SageAttention2` against baselines using head-dim=64 and headdim=128, both with and without Causal Mask (Vaswani, 2017). Detailed setup can be found

*Table 4.* **Average accuracy** across all layers of `CogvideoX` using different smoothing methods.

| Method | CosSim ↑ | Relative L1 ↓ | RMSE ↓ |
|---|---|---|---|
| None | 80.04% | 0.3906 | 0.2223 |
| HadmdAttn | 79.77% | 0.3782 | 0.2180 |
| SmoothAttn | 90.21% | 0.3383 | 0.1952 |
| Smooth K | 98.07% | 0.1493 | 0.0743 |
| Smooth Q | 98.30% | 0.1250 | 0.0712 |
| Smooth Q+K | **99.46%** | **0.0648** | **0.0334** |

*Table 5.* End-to-end metrics comparison, where $Q, K$ are quantized into INT4, while $\widetilde{P}, V$ stay in full precision.

| Q, K | Smooth (Q+K) | Llama3.1 (Lambda) ↑ | Llama3.1 (WikiText) ↓ | CogVideoX (vqa-t) ↑ |
|---|---|---|---|---|
| Full-Precision | - | 81.5% | 6.013 | 75.360 |
| INT4 | ✗ | 72.6% | 11.698 | 24.670 |
| Quantization | ✓ | **80.8%** | **6.219** | **75.147** |

*Table 6.* **Average accuracy** across all layers of `CogvideoX` using different quantization granularities.

| Method | Cos Sim ↑ | Relative L1 ↓ | RMSE ↓ |
|---|---|---|---|
| Per-token | 99.45% | 0.0649 | 0.0335 |
| **Per-thread** | **99.45%** | **0.0622** | **0.0313** |
| Per-block | 98.03% | 0.1492 | 0.0744 |
| Per-tensor | 97.15% | 0.1800 | 0.0865 |

*Table 7.* **Average accuracy** using different data types of $(\widetilde{P}, V)$ across all layers of `CogvideoX`, where $(Q, K)$ are smoothed.

| Q, K | $\widetilde{P}, V$ | Cos Sim ↑ | Relative L1 ↓ | RMSE ↓ |
|---|---|---|---|---|
| INT4 | INT8 | 77.05% | 0.5618 | 0.5044 |
| | E5M2 | 99.20% | 0.0905 | 0.0903 |
| | **E4M3** | **99.44%** | **0.0683** | **0.0347** |
| | **FP16** | 99.45% | 0.0649 | 0.0335 |

*Table 8.* End-to-end generation latency using `SageAttention2` (The latency of `Llama3.1` is the time to first token generation using different sequence lengths).

| Model | GPU | Original | SageAttn 2-8b | SageAttn 2-4b |
|---|---|---|---|---|
| CogvideoX (2B) | RTX4090 | 86 s | **54 s** | **52 s** |
| CogvideoX (1.5-5B) | RTX4090 | 1040 s | **577 s** | **555 s** |
| HunyuanVideo | L20 | 2221 s | **1486 s** | **1435 s** |
| Mochi | L20 | 2336 s | **1316 s** | **1190 s** |
| Llama3.1 (48K token) | RTX4090 | 9.2 s | **5.7 s** | **5.6 s** |
| Llama3.1 (100K token) | L20 | 39.9 s | **25.4 s** | **23.2 s** |

in Appendix A.8. Specifically, Fig. 5 shows the speed across varying sequence lengths on RTX4090, indicating that `SageAttn2-4b` and `SageAttn2-8b` are approximately 3x and 2.7x faster than FlashAttention2, and about 4.5x and 4x faster than xformers, respectively. Fig. 10, 11, 12, 13, 14, 15, and 16 in Appendix A.2 show more speed results on RTX4090, L20, H20, H100 GPUs.

**Accuracy.** Table 4 and 17 show the average accuracy of different methods with INT4 $Q, K$ and FP8 $P, V$ across all layers of `CogvideoX`. The results indicate the accuracy of `SageAttn2-4b` is superior to other baselines.

### 4.3. End-to-end Performance

**Metrics loss.** We assessed the end-to-end metrics of various models using `SageAttention2` compared to baselines. Detailed evaluation results are presented in Table 2. The results indicate that `SageAttn2-4b` outperforms all baselines and maintains most of the end-to-end accuracy across all models. Additionally, `SageAttn2-8b` incurs almost no metrics loss across various models. More experiment results on other models are shown in Appendix A.9.

**Visible image and video examples.** Fig. 6, 7, 8, and 9 show some visible comparison examples from `HunyuanVideo`, `Mochi` and `CogvideoX`. We can observe that `SageAttn2-8b` does not introduce any visible differences compared to full-precision attention, whereas `SageAttn2-4b` has minor differences but is much better than the baselines.

**End-to-end speedup.** We compared the original generation latency and the latency using `SageAttention2` for models with long sequence lengths in Table 8, observing significant speedup effects. For instance, `SageAttention2` achieved a **1.8x** speedup in `CogvideoX` (1.5-5B) without any metrics loss (`SageAttn2-8b`). `SageAttn2-4b` further accelerated these models but with a little metrics loss.

### 4.4. Ablation Study

**Overhead of techniques we proposed.** As shown in Table 18, the overhead on kernel speed of per-thread quantization, smoothing Q, and two-level accumulation are 0.35%, 3.7%, and 0% compared to the attention kernel.

**Benefit of smoothing V.** The experiment showing the benefit of smoothing $V$ is shown in Appendix. A.4.

## 5. Conclusion

We introduce `SageAttention2`, an efficient and accurate quantized attention. First, we propose to quantize matrices $(Q, K)$ in a thread-level granularity and $(\widetilde{P}, V)$ to FP8. Second, we propose a method to smooth $Q$, enhancing the accuracy of $QK^\top$. Third, we propose a two-level accumulation strategy to enhance the accuracy of FP8 $\widetilde{P}V$. `SageAttention2` is faster than FlashAttention2 and xformers by approximately **3x** and **4.5x**, respectively. Moreover, `SageAttention2` matches the speed of FlashAttention3(fp8) on the Hopper GPUs, but offers significantly higher accuracy. Extensive experiments confirm that our approach maintains end-to-end metrics across language, image, and video generation models.

# Acknowledgment

This work was supported by the NSFC Projects (Nos. 92270001, 62376131). J.Z is also supported by the XPlorer Prize.

# Impact Statement

This paper presents work that aims to advance the field of Machine Learning. There are many potential societal consequences of our work, none of which we feel must be specifically highlighted here.

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

# A. Appendix

## A.1. Visible Comparison Exmaples

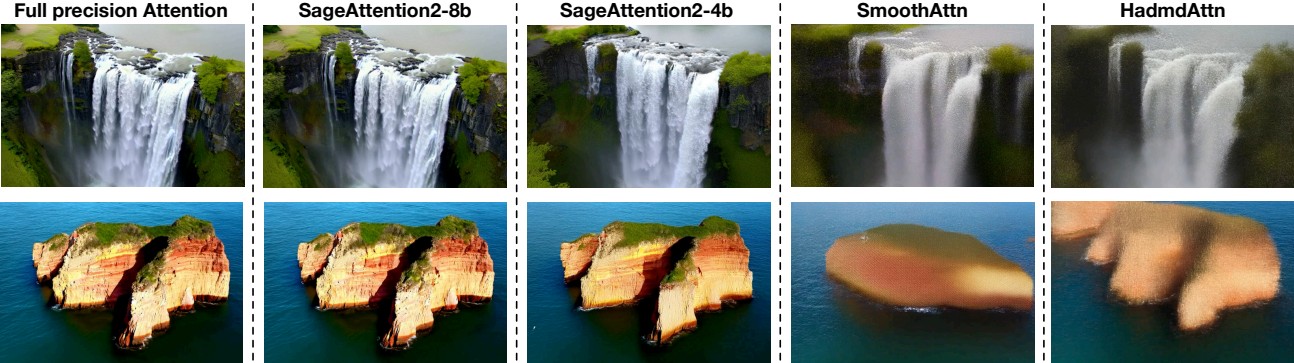

*Figure 8.* Comparison examples from `CogvideoX` (2B), prompts are sampled from open-sora prompt sets.

## A.2. Additional Kernel Speed Comparison

Fig. 10, 11, 12, 13, 14, 15, and 16 compare the speed of `SageAttention2` against baselines using headdim=64 and headdim=128, both with and without Causal Mask (Vaswani, 2017), on RTX4090, L20, H100, and H20 GPUs.

Table 9 summarizes the performance gain of different attention methods against baselines on various modern GPUs.

*Table 9.* Speedup of different attention methods on various GPUs.

| Method | 3090 | 4090 | A100 | L40 | L20 | H100 | H20 |
|---|---|---|---|---|---|---|---|
| FlashAttention2 | 1.00 | 1.00 | 1.00 | 1.00 | 1.00 | 1.00 | 1.00 |
| FlashAttention3 | ✗ | ✗ | ✗ | ✗ | ✗ | 1.37 | 1.57 |
| FlashAttention3 (fp8) | ✗ | ✗ | ✗ | ✗ | ✗ | 2.63 | 3.06 |
| SageAttention1 | 1.97 | 1.96 | 1.37 | 1.45 | 1.24 | 1.53 | 1.52 |
| SageAttention2 | ✗ | 2.93 | ✗ | 2.60 | 2.46 | 2.61 | 3.12 |

*Table 10.* An accuracy example on real tensors of `CogvideoX` model with or without smoothing $V$.

| Smooth V | Cos Sim ↑ | Relative L1 ↓ | RMSE ↓ |
|---|---|---|---|
| ✗ | 98.25% | 0.1980 | 0.2387 |
| ✓ | **99.75%** | **0.0406** | **0.0773** |

## A.3. Smoothing V

As shown in Fig. 17, this strategy could enhance the accuracy of FP22 for values in $\widetilde{P}V$ for the following reasons: Each row of $\widetilde{P}$ spans a value range from 0 to 1, and each column of $V$ in some models consistently features channel-wise biases that are exclusively positive or negative, for instance, ranging between 8 and 9 in `CogvideoX`. Consequently, the values of $\widetilde{P}V$ could be quite large. However, the floating-point number representation range is not uniform—it is denser near zero. Therefore, by subtracting the mean $\overrightarrow{V_m}$ along the channel dimension from $V$, the values of $\widetilde{P}V$ will be closer to zero, resulting in a higher representational precision (see Fig. 17 for a visual demonstration). Additionally, to maintain the correctness of the attention computation, it is only necessary to add $\overrightarrow{V_m}$ to the final calculation of $O$: $O = O + \overrightarrow{V_m}$. This is because the sum of each row of the $\widetilde{P}$ matrix equals 1, so $\widetilde{P}\overrightarrow{V_m} = \overrightarrow{V_m}$. In other words, this method decomposes $V$ into two parts: $\overrightarrow{V_m}$ and $V$. For $V$, it centers the values of each column around zero, which leads to the dot product result between a row from the quantized $\widetilde{P}$ matrix and a column from the quantized $V$ matrix being closer to zero. This makes the representation of FP22 more accurate. Meanwhile, $\overrightarrow{V_m}$ is retained in FP16 and is added to $O$ at the end without causing a loss of precision for the $\overrightarrow{V_m}$ part.

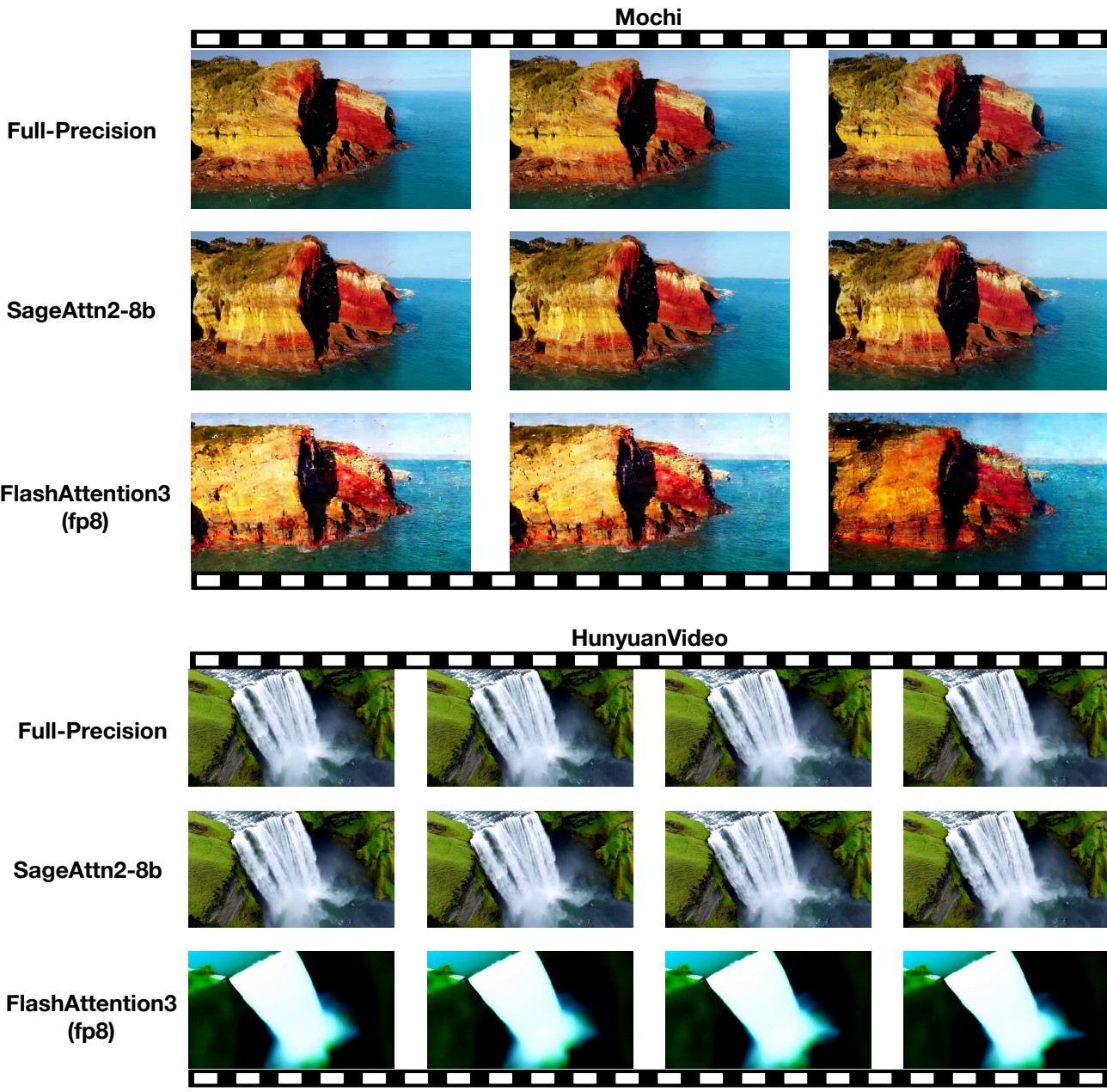

*Figure 9.* A comparison example using SageAttn2-8b and FlashAttention3 on `Mochi` and `HunyuanVideo`.

### A.4. Experiment of Smoothing V

Table 10 shows the attention accuracy on real tensors sampled from `CogvideoX` with and without smoothing $V$. It demonstrates that smoothing $V$ could improve the accuracy of `SageAttention2` when quantizing $Q, K$ to INT4 and $\widetilde{P}, V$ to FP8. We find that smoothing V is generally effective for diffusion models (Zheng et al., 2023; 2024b;a; 2025; Zhao et al., 2024; 2025a; Wang et al., 2024).

### A.5. Theoretical Analysis of Smoothing

In this section, we analyze the benefit of smoothing from a theoretical perspective. Let $X \in \mathbb{R}^{N \times d}$ be $N$ activation tokens of dimension $d$. Following (Dettmers et al., 2023), we suppose that an activation token follows an Gaussian distribution

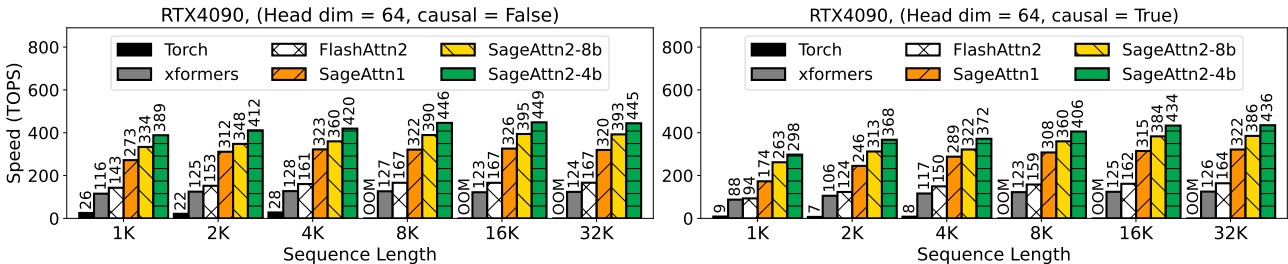

*Figure 10.* Speed comparison between `SageAttention2` and baselines (RTX4090, headdim=64).

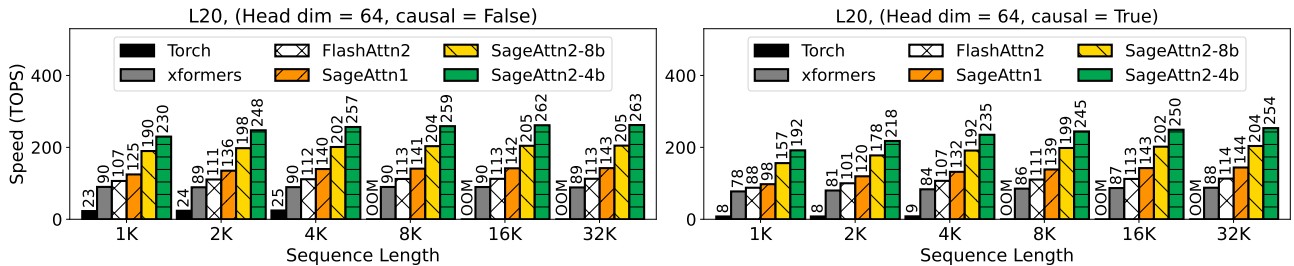

*Figure 11.* Speed comparison between `SageAttention2` and baselines (L20, headdim=64).

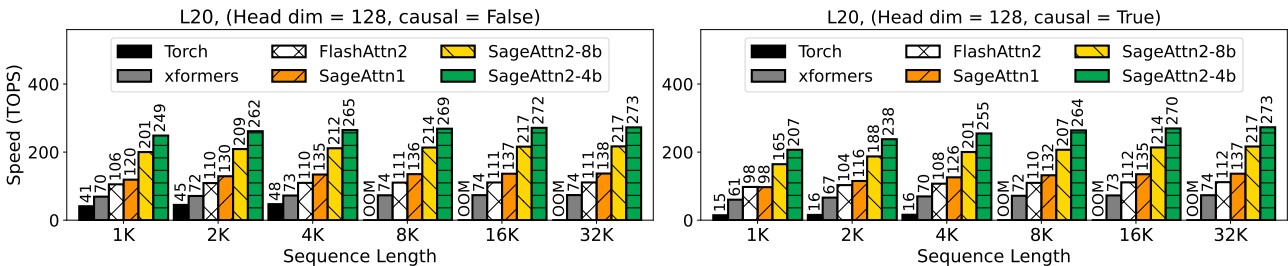

*Figure 12.* Speed comparison between `SageAttention2` and baselines (L20, headdim=128).

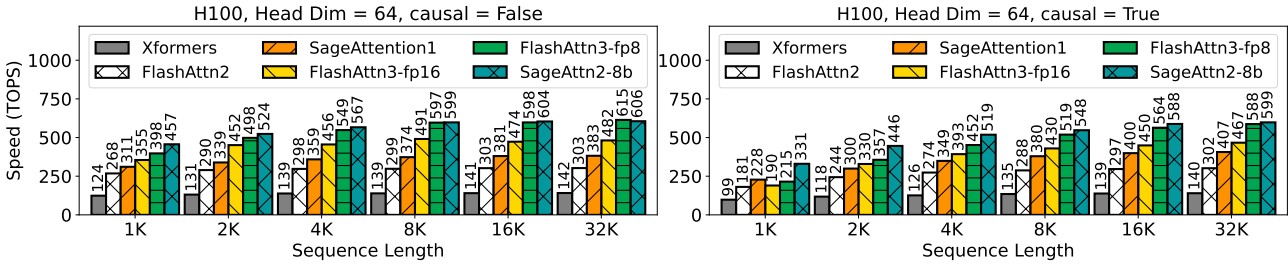

*Figure 13.* Speed comparison between `SageAttention2` and baselines (H100, headdim=64).

$\mathcal{N}(\boldsymbol{\mu}, \Sigma^2)$, where $\boldsymbol{\mu} = (\mu_1, \mu_2, \ldots, \mu_d)$ and $\Sigma^2$ is a diagonal matrix with $\Sigma^2 = \mathrm{diag}(\sigma_1^2, \sigma_2^2, \ldots, \sigma_d^2)$. Further, we suppose that different token $X_i$ is i.i.d. sampled from the same distribution.

Suppose the absolute maximum value in a quantization group (Hu et al., 2025; Zhang et al., 2025j) is $M$, and the bit width is $b$, then there are $2^b$ quantization levels. Under the round-to-nearest strategy, the expected quantization error is $\frac{1}{2} \cdot \frac{2M}{2^b}$, which is proportional to the maximum absolute value in the quantization group. So a smaller absolute maximum value leads to smaller quantization error.

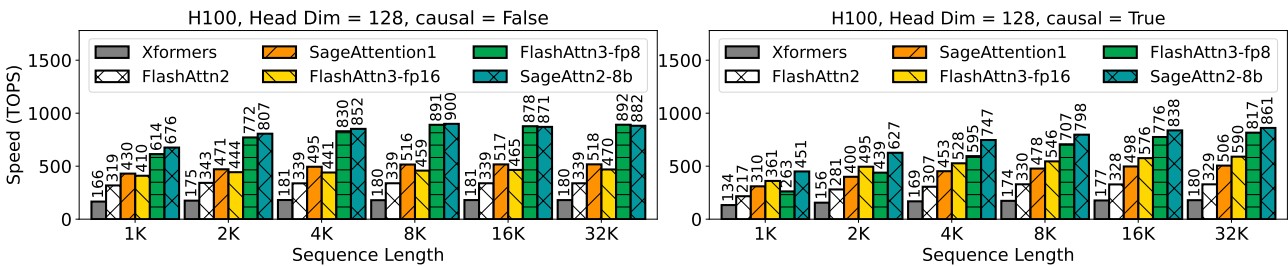

*Figure 14.* Speed comparison between `SageAttention2` and baselines (H100, headdim=128).

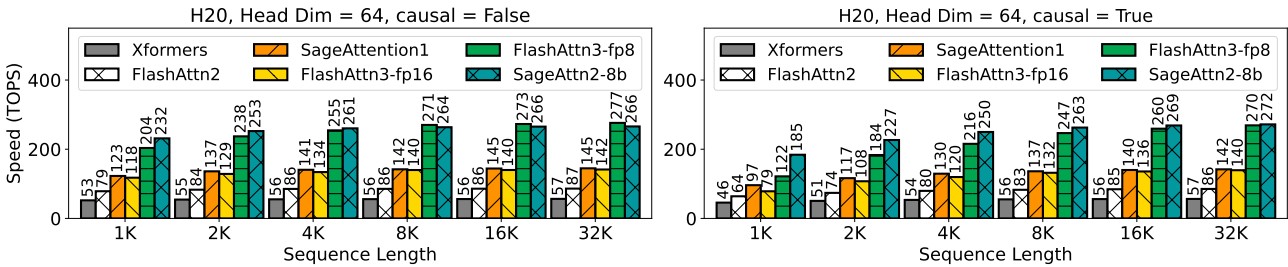

*Figure 15.* Speed comparison between `SageAttention2` and baselines (H20, headdim=64).

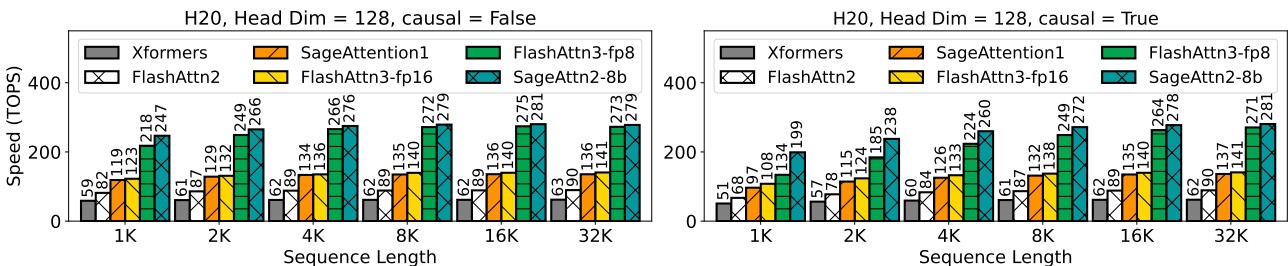

*Figure 16.* Speed comparison between `SageAttention2` and baselines (H20, headdim=128).

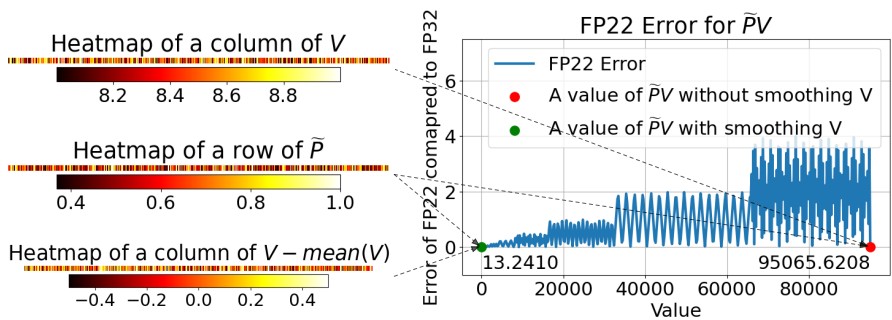

*Figure 17.* An example of dot product precison a row of $\widetilde{P}$ and a column of $V$ presented by FP22 data type.

After smoothing, we have:

$$Y_{ij} = X_{ij} - \frac{1}{N} \sum_{k=1}^{N} X_{kj} \qquad (3)$$

and $Y_{ij}$ also follows a Gaussian distribution. The mean and variance of $Y_{ij}$ can be calculated as follows:

$$\mathbb{E}[Y_{ij}] = \mathbb{E}[X_{ij}] - \frac{1}{N}\sum_{k=1}^{N}\mathbb{E}[X_{kj}] = \mu_j - \frac{1}{N}\sum_{k=1}^{N}\mu_j = 0 \tag{4}$$

$$\text{Var}[Y_{ij}] = \text{Var}[\frac{N-1}{N}X_{ij}] + \sum_{k=1,k\neq i}^{N}\text{Var}[\frac{1}{N}X_{kj}] = \frac{(N-1)^2}{N^2}\sigma_j^2 + (N-1)\frac{1}{N^2}\sigma_j^2 = \frac{(N-1)}{N}\sigma_j^2 \tag{5}$$

So $Y_{ij}$ has a smaller mean and variance compared to $X_{ij}$. Following the property of Gaussian distribution, we have:

$$P(|Y_{ij}| < \epsilon) > P(|X_{ij}| > \epsilon), \ \forall \epsilon > 0 \tag{6}$$

So the distribution of $X_{ij}$ is more concentrated toward 0 after smoothing. Then we know that

$$P(\text{absmax}(Y_i) < \epsilon) = \prod_{j=1}^{d}P(|Y_{ij}| < \epsilon) > \prod_{j=1}^{d}P(|X_{ij}| > \epsilon) = P(\text{absmax}(X_i) < \epsilon) \tag{7}$$

So this makes the distribution of absolute max value in a token more concentrated towards 0, leading to smaller quantization error.

### A.6. Per-Thread Quantization Formulation

| R\C | 0 | 1 | 2 | 3 | 4 | 5 | 6 | 7 |
|-----|---|---|---|---|---|---|---|---|
| 0 | T0: {c0, c1} | | T1: {c0, c1} | | T2: {c0, c1} | | T3: {c0, c1} | |
| 1 | T4: {c0, c1} | | T5: {c0, c1} | | T6: {c0, c1} | | T7: {c0, c1} | |
| 2 | | | | | | | | |
| | | | | | | | | |
| 7 | T28: {c0, c1} | | T29: {c0, c1} | | T30: {c0, c1} | | T31: {c0, c1} | |
| 8 | T0: {c2, c3} | | T1: {c2, c3} | | T2: {c2, c3} | | T3: {c2, c3} | |
| 9 | T4: {c2, c3} | | T5: {c2, c3} | | T6: {c2, c3} | | T7: {c2, c3} | |
| 10 | | | | | | | | |
| .. | | | | | | | | |
| 15 | T28: {c2, c3} | | T29: {c2, c3} | | T30: {c2, c3} | | T31: {c2, c3} | |

*Figure 18.* Memory layout of INT4/INT8 tensor core for accumulator matrix $C$ and $D$ in $D = A * B + C$ among 32 threads (T0 $\sim$ T31) in a warp. $C$ and $D$ is of shape 16x8. Each thread only holds 4 out of the 128 elements.

To further clarify the per-thread quantization, we first introduce the INT4 MMA instruction of Tensor Core, and then give the formulation of per-thread quantization.

Tensor cores, first introduced in NVIDIA's Volta architecture, are specialized units designed for efficient matrix-multiply-and-accumulate (MMA) operations. Their usage significantly enhances computational efficiency and performance in AI and high-performance computing (HPC) workloads. Tensor cores compute small tiles of MMA operations, specifically $D = A * B + C$ on a warp (32 contiguous threads) basis. Each thread in the warp holds a fragment of input matrices and will get a fragment of output matrix as a computation result. The INT4 `mma.m16n8k64` tensor core operation computes the product of a $16 \times 64$ INT4 matrix $A$ and a $64 \times 8$ INT4 matrix $B$, both stored in registers. It accumulates the result into a $16 \times 8$ INT32 matrix $C$, also stored in registers, and returns the final product matrix $D$, which has the same shape $(16 \times 8)$, data type (INT32), and storage location (registers). Each thread holds only $\frac{1}{32}$ of the input and output data. Fig. 18 extracted from the PTX document (NVIDIA) shows the memory layout of matrix $C$ and $D$ among 32 threads in a warp. Each thread only holds 4 out of the 128 result elements.

$$i_{\delta q} = \lfloor (n * 8 * c_w / b_q) \rfloor$$

$$q_i[i_{\delta q}] = \{8 \times (n\%8) + \lfloor (n * \frac{c_w}{b_q}) \rfloor * \frac{b_q}{c_w}\}, \; n \in [0, N]$$

$$\delta_Q[i_{\delta q}] = \frac{\max(| Q[q_i[i_{\delta q}]] |)}{7}$$

$$\hat{Q}[q_i[i_{\delta q}]] = \left\lceil \frac{Q[q_i[i_{\delta q}]]}{\delta_Q[i_{\delta q}]} \right\rceil$$

$$i_{\delta k} = \lfloor (n * 4 / b_k) \rfloor \qquad (8)$$

$$k_n[i_{\delta k}] = \{8 \times (n\%8) + \lfloor n/b_k \rfloor * b_k\} \cup$$

$$\{8 \times (n\%8) + 1 + \lfloor n/b_k \rfloor * b_k\}, \; n \in [0, N]$$

$$\delta_K[i_{\delta k}] = \frac{\max(| K[k_n[i_{\delta k}]] |)}{7}$$

$$\hat{K}[k_n[i_{\delta k}]] = \left\lceil \frac{K[k_n[i_{\delta k}]]}{\delta_K[i_{\delta k}]} \right\rceil$$

By ensuring results held by each thread share a common dequantization scale (belong to the same quantization group), we can avoid the overhead associated with per-token quantization. Leveraging this observation, we design per-thread quantization as formulated in Eq. 8, where $c_w$ is the count of GPU Warps, $b_q$ and $b_k$ are the block size of $Q, K$, and $n$ is the token index of $Q, K$. For typical block size of $b_q = 128$, $b_k = 64$ and warp number $c_w = 4$ (as used in FlashAttention2), each warp processes a tile of 32 query tokens and 64 key tokens. Query tokens $i, 8 + i, 16 + i, 24 + i$ ($i = 0, 1, \cdots, 7$) can be made into one quantization group and key tokens $j, 1 + j, 8 + j, 9 + j, \cdots, 56 + j, 57 + j$ ($j = 0, 1, 2, 3$) can be made into one quantization group, as visualized in Fig. 4. This design aligns with the memory layout of output matrix $D$ of tensor core shown in Fig. 18, ensuring that each thread only needs one $Q$ scale and one $K$ scale for dequantization.

As a result, this approach creates 32 quantization groups for $Q$ (8 for each of the 4 warps) and 4 quantization groups for $K$ in a 128x64 block, providing $32\times$ and $4\times$ finer granularity compared to per-block quantization for query tokens and key tokens, respectively. Table 6 and Table 15 show the accuracy gains by using per-thread quantization. Per-thread quantization achieves accuracy that closely matches per-token quantization, without introducing any kernel speed degradation (see Table 18 and 19).

### A.7. Datasets and Metrics in Experiments

**Datasets.** Text-to-text models are evaluated on four zero-shot tasks: WikiText (Merity et al., 2022) to assess the model's prediction confidence, LAMBADA (Paperno et al., 2016) evaluate contextual understanding, MMLU (Hendrycks et al., 2021b) for measuring knowledge across various subjects, and Longbench (Bai et al., 2024) for comprehensive assessment of long context understanding capabilities. Text-to-video models are evaluated using the open-sora (Zheng et al., 2024c) prompt sets. Text-to-image models are assessed on MJHQ-30K (Li et al., 2024). TIMM is evaluated on on three image datasets: ImageNet (Deng et al., 2009), ImageNet-Sketch (Sketch) (Wang et al., 2019), and ImageNet-Rendition (ImageNet-r) (Hendrycks et al., 2021a).

**End-to-end metrics.** For text-to-text models, we use perplexity (ppl.) (Jelinek et al., 1977) for WikiText, Accuracy (Acc.) for LAMBADA and MMLU, and Longbench score (Bai et al., 2024). For text-to-video models, following Zhao et al. (2025b), we evaluate the quality of generated videos on five metrics: CLIPSIM and CLIP-Temp (CLIP-T) (Liu et al., 2024) to measure the text-video alignment; (VQA-a) and (VQA-t) to assess the video aesthetic and technical quality, respectively; and Flow-score (FScore) for temporal consistency (Wu et al., 2023). For text-to-image models, generated images are compared with the images in MJHQ-30K dataset in three aspects: FID (Heusel et al., 2017) and sFID (Salimans et al., 2016) for fidelity evaluation, *Clipscore* (CLIP) (Hessel et al., 2021) for text-image alignment, and *ImageReward* (IR) (Xu et al., 2023) for human preference. For TIMM, we use classification accuracy.

**Accuracy metrics.** We use three metrics to assess the accuracy of quantized attention output $O'$ compared to attention output in full-precision $O$: First, we flatten $O'$ and $O$ into vectors in the shape of $1 \times n$. Then, Cosine similarity: $CosSim = \sum OO' / \sqrt{\sum O^2} \sqrt{\sum O'^2}$, Relative L1 distance: $L1 = \sum |O - O'| / \sum |O|$, Root mean square error:

$RMSE = \sqrt{(1/n) \sum (O - O')^2}$.

## A.8. Kernel Benchmark Setup

We benchmark kernel speed with a batch size of 4 and 32 attention heads across a variety of sequence lengths. Benchmarks are conducted using head dimensions of 64 and 128, both with and without Causal Mask (Vaswani, 2017). To generate input tensors for benchmarking, we follow standard practices adopted in prior works such as FlashAttention (Dao et al., 2022). For floating-point data types, inputs are drawn from a Gaussian distribution with mean 0 and standard deviation 1, while for integer data types, inputs are uniformly sampled within the representation range:[-128, 127] for INT8 and [-8, 7] for INT4.

*Table 11.* End-to-end metrics on `Llama2` (7B).

| Model | Attention | WikiText (Ppl.) ↓ | Lambda (Acc.) ↑ | MMLU (Acc.) ↑ |
|---|---|---|---|---|
| Llama2 | Full-Precision | 5.823 | 0.886 | 0.439 |
| | HadmdAttn | 6.771 | 0.860 | 0.360 |
| | SmoothAttn | 6.717 | 0.867 | 0.392 |
| | SageAttention | **5.824** | **0.887** | **0.439** |
| | SageAttn2-4b | **5.912** | **0.881** | **0.428** |
| | SageAttn2-8b | **5.828** | **0.886** | **0.438** |

*Table 12.* End-to-end metrics on `CogvideoX` (2B).

| Model | Attention | CLIPSIM ↑ | CLIP-T ↑ | VQA-a ↑ | VQA-t ↑ | FScore ↑ |
|---|---|---|---|---|---|---|
| CogvideoX (2B) | Full-Precision | 0.1836 | 0.9975 | 77.605 | 75.360 | 3.006 |
| | HadmdAttn | 0.1742 | 0.9877 | 29.780 | 23.985 | 0.499 |
| | SmoothAttn | 0.1741 | 0.9870 | 41.703 | 47.043 | 0.624 |
| | SageAttention | **0.1833** | **0.9976** | **76.997** | **71.360** | **2.988** |
| | SageAttn2-4b | **0.1821** | **0.9973** | **77.368** | **74.906** | **2.603** |
| | SageAttn2-8b | **0.1829** | **0.9977** | **76.532** | **74.281** | **2.941** |

*Table 13.* End-to-end metrics on an image classification model.

| Model | Attention | ImageNet (Acc.) ↑ | Sketch (Acc.) ↑ | ImageNet-r (Acc.) ↑ |
|---|---|---|---|---|
| TIMM | Full-Precision | 84.79% | 45.32% | 59.55% |
| | HadmdAttn | 84.50% | 44.89% | 58.80% |
| | SmoothAttn | 84.40% | 44.68% | 58.73% |
| | SageAttention | **84.74%** | **45.38%** | **59.95%** |
| | SageAttn2-4b | **86.67%** | **45.24%** | **59.29%** |
| | SageAttn2-8b | **84.79%** | **45.39%** | **59.57%** |

*Table 14.* Comparison with FlashAttention3(fp8) on `Llama-3-262k` (8B) on InfiniBench (Zhang et al., 2024) (H100 GPU).

| Attention | Eng.Sum | Eng.QA | Eng.MC | Code.Debug | Math.Find | Retr.PassKey | Retr.Num | Retr.KV | Avg. |
|---|---|---|---|---|---|---|---|---|---|
| Full-Precision | 18.03 | 12.5 | 64.19 | 24.37 | 18.29 | 100.0 | 100.0 | 7.0 | 43.05 |
| FlashAttn3-fp8 | **19.03** | 11.73 | 55.90 | 22.59 | **22.57** | **100.0** | **100.0** | 0.4 | 41.53 |
| SageAttention2 | 18.17 | **12.46** | **64.19** | **25.63** | 17.43 | **100.0** | **100.0** | 6.6 | **43.06** |

*Table 15.* **Worst accuracy** across all layers of `CogvideoX` using different quantization granularities.

| Method | Cos Sim ↑ | Relative L1 ↓ | RMSE ↓ |
|---|---|---|---|
| Per-token | 96.76% | 0.1916 | 0.0775 |
| Per-thread | 96.72% | 0.1932 | 0.0776 |
| Per-block | 90.68% | 0.3615 | 0.1490 |
| Per-tensor | 85.85% | 0.4687 | 0.2261 |

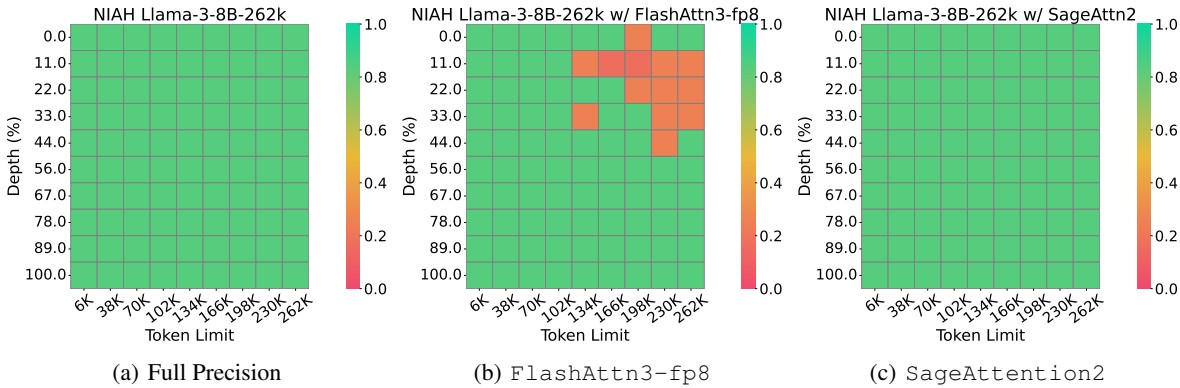

*Figure 19.* Needle In A Haystack results on `Llama-3-262k` (8B).

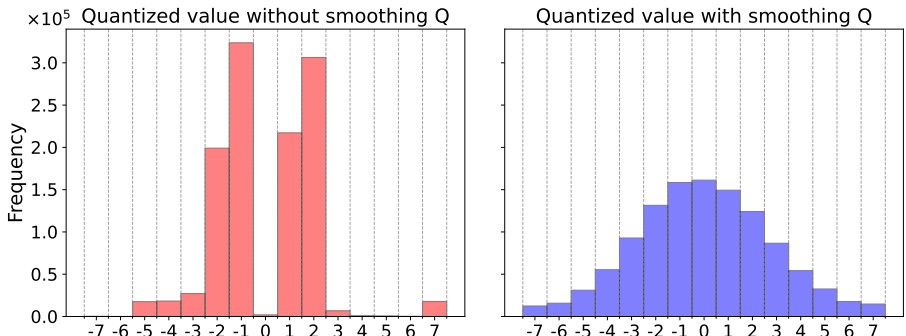

*Figure 20.* An example of quantized value distribution of $Q$ before and after smoothing $Q$.

*Table 16.* **Worst accuracy** using different data types of $(\widetilde{P}, V)$ across all layers of a `CogvideoX` model, where $(Q, K)$ are smoothed.

| $Q, K$ | $\widetilde{P}, V$ | **Cos Sim** ↑ | **Relative L1** ↓ | **RMSE** ↓ |
|---|---|---|---|---|
| INT4 | INT8 | 19.52% | 0.9579 | 1.4483 |
| | E5M2 | 94.94% | 0.2327 | 0.2361 |
| | **E4M3** | **96.70%** | **0.1956** | **0.0779** |
| | **FP16** | 96.76% | 0.1916 | 0.0775 |

*Table 17.* **Worst accuracy** across all layers of `CogvideoX` using different smooth methods.

| **Method** | **CosSim** ↑ | **Relative L1** ↓ | **RMSE** ↓ |
|---|---|---|---|
| None | 4.83% | 0.9979 | 0.7784 |
| `HadmdAttn` | 4.85% | 0.9978 | 0.7785 |
| `SmoothAttn` | 64.49% | 0.9262 | 0.7204 |
| `Smooth K` | 90.86% | 0.3565 | 0.1464 |
| `Smooth Q` | 93.10% | 0.2989 | 0.2195 |
| `SageAttn2-4b` | **96.71%** | **0.1956** | **0.0779** |

## A.9. Additional Experiments and Analysis

**Additional Results.** Table 11, 12 and 13 show results of `SageAttention2` and other baselines on `Llama2` (7B), `CogvideoX` (2B) and `TIMM`.

**Results of Super-Long Context.** We further conduct experiments on super-long context using `Llama-3-262k` (8B)[1] on

---

[1] `https://huggingface.co/gradientai/Llama-3-8B-Instruct-262k`

*Table 18.* Overhead of per-thread quantization, smoothing Q, and two-level accumulation techniques measured on L20 GPU.

| Method | TOPS |
|---|---|
| Attention (INT4 + FP8) | 284 |
| + Per-thread quantization | 283 |
| + Two-level accumulation | 283 |
| + Smoothing Q | 273 |

*Table 19.* Comparison of different quantization granularities measured on L20 GPU, with $QK^\top$ in INT4 and $\widetilde{P}V$ in FP8.

| Granularity | TOPS |
|---|---|
| Per-tensor | 286 |
| Per-block | 284 |
| Per-thread | 283 |
| Per-token | 268 |

InfiniBench (Zhang et al., 2024) and Needle-in-a-Haystack (NIAH) (Kamradt, 2023), with sequence lengths reaching up to 262k tokens on an H100 GPU. Since Hopper GPUs lack native INT4 tensor core support, we use `SageAttention2-8b` for this evaluation. We compare it against FlashAttention3(fp8), ensuring both methods operate under the same bit width. Results are shown in Table 14 and Fig 19. `SageAttention2` maintains model performance even under super-long context, while FlashAttention3(fp8) suffers from end-to-end accuracy degradation.

**Results of Audio Tasks.** We evaluate `Qwen2-Audio` (7b) (Chu et al., 2024), a speech-to-text model, on the ASR task using the Librispeech (Panayotov et al., 2015) test split and measured its performance with the WER metric (Word Error Rate). As shown in Table 20, `SageAttention2` consistently outperforms the baselines, highlighting its effectiveness in audio-related models and benchmarks.

*Table 20.* End-to-end metrics on `Qwen2-Audio` (7B).

| Model | Attention | Test-Clean ↓ | Test-Dev ↓ |
|---|---|---|---|
| | Full-Precision | 1.74 | 4.01 |
| | `HadmdAttn` | 1.77 | 4.05 |
| | `SmoothAttn` | 1.76 | 4.01 |
| `Qwen2-Audio` | `SageAttention` | **1.74** | **4.02** |
| | `SageAttn2-4b` | **1.73** | **3.99** |
| | `SageAttn2-8b` | **1.72** | **4.03** |

