# OpenReview forum: "SageAttention2: Efficient Attention with Thorough Outlier Smoothing and Per-thread INT4 Quantization"
_ICML.cc/2025/Conference — ICML 2025 poster_

### Official Review · Reviewer_aDo7 · 2025-03-11

**Overall Recommendation:** 3

**Summary:**

In Transformer-based models, SageAttention accelerates self-attention through quantization, but its use of INT8 for queries and keys is slower than INT4. Moreover, its acceleration is limited to specific Nvidia architectures due to FP16 computations. To address this issue, this paper proposes a thread-level granularity quantization method that considers GPU architecture and applies outlier smoothing to queries. Furthermore, the study identifies architectural issues in FP16 matrix multiplication accumulation operations and introduces two-level accumulation strategies to mitigate these problems. As a result, SageAttention2 achieves superior performance and acceleration compared to various existing attention mechanisms, including FlashAttention, across diverse models and tasks.

##  update after rebuttal
Thank you for the authors' response. After careful review, I believe the experiments and responses are sufficient. Therefore, I have revised the score for this paper.

**Claims And Evidence:**

The proposed method effectively leverages low-bit operations for acceleration, demonstrating its suitability for efficient computation. Additionally, it visualizes and compares the data distribution of queries, keys, and values, providing a solid rationale for determining the granularity of each quantization. Through these analyses, this paper further demonstrates that the proposed model achieves high acceleration while maintaining reasonable performance, as evidenced by comparative examples of actual image generation.

**Essential References Not Discussed:**

All essential references have been included.

**Experimental Designs Or Analyses:**

The experiments are valid, as SageAttention2 has been compared across various models and tasks against multiple existing attention mechanisms.

**Methods And Evaluation Criteria:**

The proposed methods and evaluation criteria are appropriate for the problem at hand. Experiments were conducted across multiple benchmarks, including text, video, and various other tasks.

**Other Comments Or Suggestions:**

• This paper compares model performance with existing techniques, including Smooth Attention and Hadamard Attention. However, it does not evaluate their inference speed, which would be a valuable addition.

• The figures lack readability and explanation. For example, the authors mention converting P and V to FP8, but Figure 2 and other visual materials do not clearly illustrate this process.

**Other Strengths And Weaknesses:**

1. Strenths

• This paper identifies and analyzes the discrepancy between the theoretical accuracy of the proposed quantization method and its actual performance during GPU inference. By addressing this issue, the study demonstrates effective model acceleration while maintaining the intended accuracy.

• This study analyzes the limitations of existing research and proposes improvements, conducting experiments on various models, including LLMs. The results demonstrate effective acceleration of model inference across a broad range of GPU environments.

2. Weaknesses

• This paper identifies errors arising during the accumulation process in matrix multiplication between the attention map and V. To mitigate this issue, it introduces a two-level accumulation technique. However, as acknowledged by the authors, this approach is not entirely novel. Additionally, the root cause of the problem stems from a design flaw in Nvidia GPU architecture. Since the proposed method primarily addresses a hardware-related limitation rather than introducing a fundamentally new algorithmic contribution, its impact on advancing the field is limited.

• The proposed method enhances performance compared to SageAttention by reducing the quantization bit and mitigating the resulting performance degradation. This is achieved by extending outlier smoothing—previously applied only to the key—to the query and modifying the quantization granularity to a thread-wise level. However, the first approach is a straightforward extension of an existing technique rather than a novel innovation. Moreover, the change in granularity does not lead to a substantial improvement over previous methods, making the overall contribution appear insufficient.

• A proper ablation study is needed. The ablation study provided in the appendix only compares TOPS, which focuses solely on computational speed. This is insufficient to fully explain the impact on actual model performance and makes it difficult to evaluate overall trade-offs.

• While the paper explains why per-thread quantization was chosen, it lacks comparisons with other quantization methods.

**Questions For Authors:**

• SageAttention2 performs well in computations with long sequences. However, can it still provide meaningful speed advantages in cases with short sequences or small batch sizes?

• In the paper, value uses channel-wise granularity, unlike query and key. This appears to be due to value being computed with FP8 precision. Are there any results comparing the effects of using different quantization granularities specifically for value?

**Relation To Broader Scientific Literature:**

This paper builds on prior research in self-attention acceleration, particularly SageAttention, which introduced quantization for improved efficiency. However, its reliance on INT8 for queries and keys and FP16 for attention maps limited acceleration benefits to specific Nvidia architectures.

By introducing thread-level granularity quantization and extending outlier smoothing to queries, this study refines existing quantization techniques, aligning with prior work on mixed-precision inference and INT4 quantization. Additionally, the identification of FP16 matrix multiplication accumulation issues and the proposed two-level accumulation strategy contribute to ongoing research on numerical precision challenges in GPU-based deep learning.

The study’s experimental validation across various models, including LLMs, reinforces the effectiveness of quantization-aware strategies in optimizing Transformer inference across different GPU architectures.

**Theoretical Claims:**

The paper does not make significant theoretical claims that require proof verification. Instead, it focuses on empirical analysis and performance evaluation rather than theoretical derivations.

---

> ### Author Rebuttal · Authors · 2025-04-01
>
> Dear Reviewer aDo7,
> Thank you for your valuable suggestions and questions.
>
> ---
> >### Weakness1
>
> **Reply**: We appreciate the valuable question. We argue that:
>
> 1. We first discover the critical role of accumulator precision in PV matrix multiplication for Attention; this is an essential insight for the attention operator.
>
> 2. The technique and the methods mentioned in the paper were developed concurrently (within a two-month interval). Additionally, we were the first to discover and report this hardware issue, and we can provide evidence after the rebuttal.
>
> 3. Beyond addressing hardware limitations, The technique can enhance accuracy for efficiency techniques using reduced accumulator precision in matrix multiplication.
>
> ---
> >### Weakness2
>
> **Reply**: Thank you for your question. We argue that:
>
> 1.  Smooth Q and Smooth K are distinct. Specifically, smooth Q is a per-block level smoothing and introduces an additional matrix-vector multiplication to compute ΔS added in the GPU kernel. Moreover, despite its simplicity, this approach demonstrates exceptional improvement. We argue that contribution should not be dismissed based on methodological simplicity.
>
> 2. The improvement from per-thread quantization over SageAttention’s per-block quantization is significant, not marginal. For example, Table 6 clearly shows superior accuracy with per-thread quantization. Also, using per-block quantization for Sage2-4b will get completely blurred videos of Cogvideo.
>
> 3. Per-thread quantization is a highly innovative approach, and is non-trivial, requiring sophisticated algorithm-hardware co-design.
>
> ---
> >### Weakness3
>
> **Reply**:
> Thank you for your valuable suggestion. We add the accuracy ablation results and end-to-end perplexity of Llama-3.1 alongside the TOPS and will revise Table 18.
>
> |Method|TOPS|Cossim↑|Relative L1↓|Ppl.↓|
> |-|-|-|-|-|
> |Attention (INT4 + FP8)|284|0.8004|0.3906|7.963|
> |+ Per-thread quantization|283|0.9249|0.3127|7.459|
> |+ Two-level accumulation|283|0.9498|0.2731|7.345|
> |+ Smooth Q|273|0.9946|0.06480|6.256|
>
> ---
> >### Weakness4
>
> **Reply**: We appreciate your suggestion about comparing other quantization methods. We want to highlight that our paper already includes comparisons with per-tensor, per-block, and per-token quantization methods in Table 6 and 15, with discussions in lines 250–256. The results demonstrate that per-thread quantization achieves accuracy comparable to per-token quantization and outperforms per-tensor and per-block quantization.
>
> Furthermore, we provide a speed comparison of different quantization methods, showing that  per-thread quantization introduces almost no computational overhead compared to per-tensor and per-block methods, while per-token quantization is notably slower:
>
> |Quantization|TOPS|
> |-|-|
> |per-tensor|286|
> |per-block (line 1094)|284|
> |per-thread (line 1095)|283|
> |per-token|268|
>
> ---
> >### Comment1
>
> **Reply**: We appreciate the reviewer’s suggestion. We evaluate the end-to-end inference latency of Mochi on L20 below:
>
> |Attention|Latency (s)|
> |-|-|
> |HadamardAttn|1198|
> |SmoothAttn|1208|
> |SageAttentoin2|1190|
>
> SageAttention2 achieves a similar inference speed to HadamardAttn and SmoothAttn while delivering significantly better accuracy performance.
>
> ---
> >### Comment2
>
> **Reply**: Thank you for your suggestion. First, V's FP8 conversion is already shown in Figure 2. Second, the P matrix only exists internally within the GPU kernel and is difficult to visually represent in the overview figure. Therefore, we mainly describe its FP8 conversion process in detail in Algorithm 1 and Section 3.3. We will revise Figure 2 in our paper to explicitly indicate that P is converted to FP8.
>
> ---
> >### Question1
>
> **Reply**: We appreciate the reviewer’s question. Our speed evaluation already covers a range of sequence lengths from 1,024 to 32,768, demonstrating speedup across all sequences (e.g., 1,024). To further address your concern, we provide additional results for batch size = 1 and sequence length = 1024 on RTX4090, as shown below:
>
> |Attention|TOPS|
> |-|-|
> |Torch|10.9|
> |xformers|94.1|
> |FlashAttn2|142.5|
> |SageAttn1|255.3|
> |SageAttn2-8b|329.9|
> |SageAttn2-4b|352.6|
>
> The results show that SageAttention2 consistently delivers higher throughput with short sequences and small batch sizes, maintaining similar speedup as observed in long-sequence and large-batch scenarios.
>
> ---
> >### Question2
>
> **Reply**: Thank you for your valuable suggestion. First, per-token quantization can not be applied to V because the quantization must be conducted along the outer axis of $PV$. We compare the accuracy of per-channel, per-block, and per-tensor quantization methods, showing that per-channel quantization achieves the best accuracy:
>
> |Quantization|Cossim↑|Relative L1↓|
> |-|-|-|
> |Per-Channel|0.9946|0.0648|
> |Per-Token|✗|✗|
> |Per-Block|0.9930|0.0651|
> |Per-Tensor|0.9922|0.06777|
>
> ---
> If you feel your concerns have been resolved, we would greatly appreciate it if you consider raising the score.

---

### Official Review · Reviewer_r8J3 · 2025-03-14

**Overall Recommendation:** 5

**Summary:**

The authors propose SageAttention2, where they manage to quantize key and query matrices in the attention computation to INT4 while the softmax outputs and value matrices are quantized to FP8. They show that the quality degradation is managable in this configuration. Meanwhile, if the key and query matrices are quantized to INT8 instead, then there is almost no measurable quality degradation. The key is to perform asymmetric quantization of the key and query matrices.

## update after rebuttal
My score is unchanged after the discussion period. Keeping my score to champion for the authors' work.

**Claims And Evidence:**

Yes.

**Essential References Not Discussed:**

No.

**Experimental Designs Or Analyses:**

Yes.

**Methods And Evaluation Criteria:**

Yes.

**Other Comments Or Suggestions:**

No other comments.

**Other Strengths And Weaknesses:**

* The paper is well written.
* The experimental design is comprehensive.
* This work will be the new state-of-the-art attention kernel.

**Questions For Authors:**

* Do the authors think that it is possible for PV computation to be quantized to sub-8-precision as well?

**Relation To Broader Scientific Literature:**

Yes. Attention is a widely used operation. This work will be impactful.

**Theoretical Claims:**

There aren't any theoretical claims except for simple mathematical derivations. They seem correct.

---

> ### Author Rebuttal · Authors · 2025-04-01
>
> Dear Reviewer r8J3,
> Thank you for your valuable question. Below, we address the question raised.
>
> ---
> >**Question1.** Do the authors think that it is possible for PV computation to be quantized to sub-8-precision as well?
>
> **Reply**:
> Thank you for the insightful suggestion. The answer is yes - we have tried quantizing P and V to FP4 using Blackwell GPUs' micro-scaling quantization and obtained preliminary accuracy results across CogVideo layers.
>
> Specifically:
> - We keep Q and K in INT8 (since Blackwell GPUs don't support INT4)
> - We quantize P and V to NVFP4
> - Using our smooth Q and smooth K techniques
>
> |Attention|Cossim ↑|Relative L1 ↓|
> |-|-|-|
> |Sage2-8b|0.99982|0.01573|
> |Sage2-4b|0.99460|0.06480|
> |FP4 PV Attention|0.99674|0.03250|
>
> Based on these preliminary results, we believe FP4 quantization for P and V is possible and practical.

---

### Official Review · Reviewer_t1YD · 2025-03-15

**Overall Recommendation:** 4

**Summary:**

This paper makes the attention computation more efficient. It uses INT4 quantization of Q and K, instead of INT8 quantization. To enhance the accuracy of INT4, this paper proposes a outlier smoothing strategy, which is well-motivated. The overall design and implementation take hardware characteristics into account. The accuracy and efficiency of proposed method are evaluated by various tasks and settings: on RTX4090, the proposed method is much faster than FlashAttention2; on Hopper GPUs, the proposed method matches the speed of FlashAttention3 while delivers much higher accuracy.

**Claims And Evidence:**

All claims are supported by clear and convincing evidence.

**Essential References Not Discussed:**

No

**Experimental Designs Or Analyses:**

I have checked all experiments represented in main text. I think the experimental designs are sound and sufficient.

1. It compares with FlashAttention and recent proposed INT4 quantizations under various tasks and settings.
2. The ablation study is sufficient to demonstrate the effectiveness of its key designs, including Q/K smoothing strategies, Q/K quantization granularities, and numerical precision choices of P/V.

**Methods And Evaluation Criteria:**

The proposed methods and evaluation criteria make sense.

**Other Comments Or Suggestions:**

The experimental settings of Figure 5 are not clear. What is number of heads? Which dataset is used?

**Other Strengths And Weaknesses:**

Strengths: The proposed takes hardware characteristics into account. The proposed INT4 Per-thread Quantization is novel and it is a great example of algorithm/hardware co-design.

Weaknesses: The effectiveness of proposed Q/K smoothing is evaluated empirically. The evaluation is sufficient but it might be better to analysis the theoretical benefits of Q/K smoothing. For example, the authors can analysis and compare the quantization error bounds with/without smoothing for specific input distributions. This provides the paper with stronger theoretical support.

**Questions For Authors:**

The authors mentioned that the accumulator for the mma(f32f8f8f32) instruction is actually FP22. I am not familiar with mma instruction. Is the precision mismatch some kind of software "bug"? Or perhaps there are deeper design reasons behind it?

**Relation To Broader Scientific Literature:**

algorithm and hardware co-design

**Theoretical Claims:**

There are no theoretical claims in this paper.

This paper does some mathematical derivations about quantization in section 3.1 and I have checked its correctness. I also have checked the implementation of SageAttention2 in algorithm 1.

---

> ### Author Rebuttal · Authors · 2025-04-01
>
> Dear Reviewer t1YD,
> Thank you for your valuable suggestions and questions. Below, we address each point raised.
>
> ---
> >**Weakness1.** The effectiveness of proposed Q/K smoothing is evaluated empirically. The evaluation is sufficient but it might be better to analysis the theoretical benefits of Q/K smoothing. For example, the authors can analysis and compare the quantization error bounds with/without smoothing for specific input distributions. This provides the paper with stronger theoretical support.
>
> **Reply**: Thank you for the valuable suggestion. We analyze the quantization error as follows:
>
> **Proof**:
> Let $X \in \mathbb{R}^{N \times d}$ be $N$ activation tokens with dimension $d$.
>
> Following [1], we suppose that an activation token follows an Gaussian distribution $\mathcal{N}(\boldsymbol{\mu}, \Sigma^2)$, where $\boldsymbol{\mu} = (\mu_1, \mu_2, \ldots, \mu_d)$ and $\Sigma^2$ is a diagonal matrix with $\Sigma^2 = \mathrm{diag}(\sigma_1^2, \sigma_2^2, \ldots, \sigma_d^2)$.
>
> Further, we suppose that different token $X_i$ is i.i.d. sampled from the same distribution.
>
> Suppose the absolute maximum value in a quantization group is $M$, and the bit width is $b$, then there are $2^b$ quantization levels. Under round to nearest strategy, the expected quantization error is $\frac{1}{2} \cdot \frac{2M}{2^b}$ which is proportional to the maximum absolute value in the quantization group. So smaller absolute maximum value leads to smaller quantization error.
>
> After smoothing, we have:
> $$
> Y_{ij} = X_{ij} -  \frac{1}{N} \sum_{k=1}^N X_{kj}
> $$
> $Y_{ij}$ also follows a Gaussian distribution. The mean and variance of $Y_{ij}$ can be calculated as follows:
>
> $$
> \mathbb{E}[Y_{ij}] = \mathbb{E}[X_{ij}] - \frac{1}{N} \sum_{k=1}^N \mathbb{E}[X_{kj}] = \mu_j - \frac{1}{N} \sum_{k=1}^N \mu_j = 0
> $$
>
> $$
> \mathrm{Var}[Y_{ij}] = \mathrm{Var}[\frac{N-1}{N} X_{ij}] + \sum_{k=1, k\neq i}^N \mathrm{Var}[\frac{1}{N} X_{kj}]
> = \frac{(N-1)^2}{N^2} \sigma_j^2 + (N-1) \frac{1}{N^2} \sigma_j^2 = \frac{(N-1)}{N} \sigma_j^2
> $$
>
> So $Y_{ij}$ have smaller mean and variance compared to $X_{ij}$. Following the property of Gaussian distribution, we have:
> $$
> P(|Y_{ij}| < \epsilon) > P(|X_{ij}| > \epsilon),\ \forall \epsilon > 0
> $$
>
> So the distribution of $X_{ij}$ is more concentrated towards 0 after smoothing. Then we know that
>
> $$
> P(\mathrm{absmax}(Y_i) < \epsilon) = \prod_{j=1}^d P(|Y_{ij}| < \epsilon) > \prod_{j=1}^d P(|X_{ij}| > \epsilon) = P(\mathrm{absmax}(X_i) < \epsilon)
> $$
>
> So this makes the distribution of absolute max value in a token more concentrated towards 0, leading to smaller quantization error.
>
> ---
> >**Comment1.** The experimental settings of Figure 5 are not clear. What is number of heads? Which dataset is used?
>
> **Reply**: We apologize for the lack of clarity in the experimental settings of Figure 5. We used a head size of 32 and a batch size of 4. Since we are benchmarking kernel speed, we follow standard practice as FlashAttention1/2/3, i.e., using Gaussian input (with mean 0, standard variance 1) for floating-point inputs. For integer inputs, we use uniform random values within the representation range: [-128, 127] for INT8 and [-8, 7] for INT4. We will clarify this in the final manuscript.
>
> ---
> >**Question1.** The authors mentioned that the accumulator for the mma(f32f8f8f32) instruction is actually FP22. I am not familiar with mma instruction. Is the precision mismatch some kind of software "bug"? Or perhaps there are deeper design reasons behind it?
>
> **Reply**: Thank you for your question. This is a hardware-level issue, not a software bug. The exact reason for this precision mismatch is not publicly documented, but we hypothesize that it is due to chip area constraints. Given the limited space, the designers may have had to make trade-offs, and reducing the accumulator precision for the FP8 tensor appears to have been one such compromise.
>
> [1] QLoRA: Efficient Finetuning of Quantized LLMs
>
> ---
> If you feel your concerns have been resolved, we would greatly appreciate it if you consider raising the score.

---

### Official Review · Reviewer_KKjK · 2025-03-17

**Overall Recommendation:** 4

**Summary:**

This paper proposed several improvements on SageAttention to make it comparable with FlashAttention 3 in terms of speed but better in accuracy. The enhancements mainly focused on enabling lower precision compared to previous SageAttention, i.e. move from INT8 to INT4 for Q*K^T and from FP16 to FP8 for P*V. To address the outlier problem in Q*K^T, SageAttention2 extended the smoothing technique used in SageAttention from K-only to both Q and K and showed great improvement in accuracy. In order to further improve accuracy and reduce the overhead from INT4 dequantization, author proposed an interesting per-thread quantization scheme which will enforce each thread to read and use only one set of Q K scale. This method shows comparable accuracy with per-token, i.e. better than per-block, and costs almost no overhead. The challenge for using FP8 in P*V calculation is caused by Hopper TensorCore design, which uses FP22 accumulator instead of FP32. To alleviate the impact of FP22, author employed the two-level accumulation technique and optionally applied smoothing on V. Resulting SageAttention2 shows very promising results on language, image, and video models.

**Claims And Evidence:**

Yes

**Essential References Not Discussed:**

No.

**Experimental Designs Or Analyses:**

Yes, no issues.

**Methods And Evaluation Criteria:**

Yes

**Other Comments Or Suggestions:**

type on Line 115/116 right column, "... preprocessing technique **to by** subtracting the token-wise mean..."

**Other Strengths And Weaknesses:**

Strength
1. describe the key concepts clearly with good supporting data.
2. tested on a wide range of models.
3. demonstrated speed-up and accuracy improvement.

Weakness
overall, a nice, solid work. Here are a few minor suggestions:
1. Author highlighted several times about the comparison with FlashAttention2/3 throughout the paper, however, FlashAttn were missing in many cases in the main experimental results, for example, only the middle part of Table 2 (for video models) shows FlashAttn3-fp8, but the other two categories only have SageAttn family. Since it was mentioned in Abstract, Introduction, and Fig. 1, readers would be expecting a quantitative benchmark with both FlashAttn2 and 3. Especially when SageAttn2 provides two modes, i.e. one is faster (INT4) while the other is more accurate (INT8), a dedicated table for a fair comparison to FlashAttn family would be beneficial.

2. In Section 3.2, author stated that per-token method would result in "significant overhead" but didn't specify/quantify how serious this problem is. On the other hand, in Table 18, it only shows per-thread method added no overhead. Maybe author can elaborate a bit in Sec 3.2 and give readers a better idea how much improvement were made by this technique.

3. In Appendix Table 9, 14, and Fig 18, it wasn't clear which version of SageAttention2 (4b or 8b) were used in the experiments.

**Questions For Authors:**

Please see Weakness above

**Relation To Broader Scientific Literature:**

It's an improvement on SageAttention and competitive alternative to FlashAttention2/3.

**Theoretical Claims:**

NA, no theoretical claims.

---

> ### Author Rebuttal · Authors · 2025-04-01
>
> Dear Reviewer KKjK,
> Thank you for your valuable suggestions and questions. Below, we address each point raised.
>
> ---
> >**W1.** Author highlighted several times about the comparison with FlashAttention2/3 throughout the paper, however, FlashAttn were missing in many cases in the main experimental results, for example, only the middle part of Table 2 (for video models) shows FlashAttn3-fp8, but the other two categories only have SageAttn family. Since it was mentioned in Abstract, Introduction, and Fig. 1, readers would be expecting a quantitative benchmark with both FlashAttn2 and 3. Especially when SageAttn2 provides two modes, i.e. one is faster (INT4) while the other is more accurate (INT8), a dedicated table for a fair comparison to FlashAttn family would be beneficial.
>
> **Reply**: We apologize for any confusion regarding the comparison with FlashAttention2/3. The *"full precision"* row in our tables represents FlashAttention2/3’s unquantized attention performance, making Table 2 a direct and fair comparison between SageAttn2 and FlashAttention2/3. Note that since FlashAttention3 can only run on Hopper GPUs (H100/H20), so the speed comparisons with it are naturally limited to these GPUs. We will revise the manuscript to explicitly clarify this points.
>
> ---
> >**W2.** In Section 3.2, author stated that per-token method would result in "significant overhead" but didn't specify/quantify how serious this problem is. On the other hand, in Table 18, it only shows per-thread method added no overhead. Maybe author can elaborate a bit in Sec 3.2 and give readers a better idea how much improvement were made by this technique.
>
> **Reply**: Thank you for your valuable feedback. We appreciate your point about the lack of clarification on the overhead introduced by the per-token method in Section 3.2.
>
> We measure the TOPS of per-token INT4 quantization and the result is as follows, with the settings aligned with Table 18:
>
> |Quantization|TOPS|
> |-|-|
> |per-block (line 1094 in our paper)|284|
> |per-thread (line 1095 in our paper)|283|
> |per-token|268|
>
> Per-thread introduces about 0.4\% overhead, while per-token quantization results in approximately 6\% overhead, which is 15 times the overhead of per-thread quantization.
>
> ---
> >**W3.** In Appendix Table 9, 14, and Fig 18, it wasn't clear which version of SageAttention2 (4b or 8b) were used in the experiments.
>
> **Reply**: We apologize for the ambiguity. On Hopper GPUs (H100/H20), INT4 tensor core is not available. So in Figure 9, the columns of H100 and H20 report the result of SageAttn2-8b, while all other entries of SageAttn2 use the 4b version. For Table 14 and Fig. 18, the experiment was conducted on H100 (as stated in line 1041), so we use SageAttn2-8b and compared it with FlashAttention3-FP8, which has the same bit width, for a fair evaluation. We will clarify this in the final manuscript.
>
> ---
> >**C1.** typo on Line 115/116 right column
>
> **Reply**: Thank you for pointing out the typo and we will revise our paper.
>
> ---
> If you feel your concerns have been resolved, we would greatly appreciate it if you consider raising the score.

---

> > ### Comment · Reviewer_KKjK · 2025-04-03
> >
> > Thanks for your clarifications. Please do include (better in earlier paragraphs) the considerations regarding INT4 Tensor Core availability on Hopper. It will remind the readers about the deployment options.

---

> > > ### Author Response · Authors · 2025-04-03
> > >
> > > Dear Reviewer KKjK,
> > >
> > >  Thank you for your valuable suggestion and timely feedback! We will revise our paper to include information about the availability of INT4 Tensor Cores on Hopper GPUs in the earlier paragraphs. Furthermore, we will also revise our paper to include clarifications on all the issues raised in the rebuttal.
> > >
> > > We hope our reply can address your concerns. We would greatly appreciate it if you consider raising the score.

---

### Official Review · Reviewer_1zQu · 2025-03-26

**Overall Recommendation:** 4

**Summary:**

SageAttention2 introduces a new way to enhance the accuracy and efficiency of attention through a 3 pronged process: firstly, it introduces an INT4 matrix-multiplication technique for query-key and FP8 matmul technique for attention weight and values; secondly, it proposes a smoothing technique for queries to reduce loss of accuracy due to outliers in INT4 quantization; thirdly, it investigates the reason for loss in accuracy due to FP8 matmul for attention weights and values, and proposes a way to overcome the loss.

The main algorithmic contributions are as follows:
1. Providing an efficient INT4 quantization scheme that adapts to thread-level granularity to move from multiple quantization scales in a GPU Warp to single quantization scale per warp to reduce dequantization overhead. This technique takes adavantage of the Nvidia's PTX Tensor Core warp-level `mma` instruction layout to partition Q and K thread blocks into quantization groups such that each quantization group shares a single quantization scaling factor to perform dequantization.

2. To reduce the accuracy overhead from loss of precision due to INT4 range, the paper adapts a technique to smooth Q similar to a previous technique of smoothing K from SageAttention paper. This technique subtracts the mean along the token dimension to utilize the INT4 range `[-7, 7]` more uniformly. This reduces the effect of outliers by making them smaller in magnitude, which in turn allows for a significant preservation of accuracy.

3. For `P*V`, the paper choses FP8 quantization over INT or FP16 due to better representation of P values numerically and prevalence of FP32 accumulator in wide variety of GPUs respectively. The authors observe that the loss in accuracy due E4M3 FP8 quantization of PV (P per-block, V per-channel quantization) is due to the nature of FP22 accumulator in the CUDA implementation, and they propose a 2 stage FP32 accumulation buffer during the computation of the online-softmax PV matmul to counteract the loss in accuracy due to the CUDA implementation using FP22 accumulator.

4. Lastly the authors propose the same smoothing technique for V by subtracting mean along the token dimension to counteract the accuracy loss due to FP22 accumulation, but its benefit is only found to be observed when V possesses channel-wise bias, so this smoothing is kept as optional.

**Claims And Evidence:**

Most of the claims made in the paper over accuracy and performance are backed by sufficient evidence.

Observations:
1. Smoothing of Q+K seems to be an effective technique in preservation of accuracy with INT4/8 matmul of QK.
2. The granular per-thread quantization seems to be on-par with per-token quantization strategy for INT4 QK matmul in terms of accuracy, though enough evidence hasn't been presented on the said overhead of per-token dequantization. Theoretically, per-thread quantization shouldn't have any performance overhead, which is what has been presented in the table 18, and evidence of it being able to handle outliers in preserving accuracy has been tabulated in Table 4.
3. The 2-level FP32 buffer used for FP22 accumulation leads to circumvent the accuracy loss due to mma(f32f8f8f32) CUDA implementation. Though this claim hasn't been backed by sufficient evidence in terms of accuracy over multiple tasks. Table 7 and 16 fail to mention if this 2 level was employed in FP8 matmul of PV.

**Essential References Not Discussed:**

Not related to understanding the topics in this paper directly, but would be nice to get a short reference and methodology discussed in FlashAttention2. FlashAttention2 also implements methodology to eliminate per-warp communication which is similar to the computation done in SageAttention2 per-thread INT4 quantization with single scaling factor to eliminate dequantization overhead.

**Experimental Designs Or Analyses:**

The paper goes over multiple experiments with regards to kernel speed comparing with xformers, SageAttention, FlashAttention2, and FlashAttention3 (for Nvidia Hopper architecture). The analysis for the experiments are found to be accurate in text with regards to TOPS. There is some room for improvement over the TOPS claim, as the end-to-end generation latency across LLMs show that switching to SageAttention2-4b over 8b might not be the best use of hardware (Table 8), given that 8b suffers almost no loss in accuracy over the full-precision attention, and has quite similar end-to-end generation latency as compared to 4b.

The paper also mentions the metric loss of SageAttention2-4b compared to 8b, which is apparent once the tabular and visual results are inspected upon.

**Methods And Evaluation Criteria:**

The paper covers many modalities across multiple tasks involving language, image, and video generation, along with image classification involving an extensive suit of models that have attention at their core for computation.
1. The SageAttention2 is benchmarked against multiple different renowned efficient attention algorithms, in text2text, text2video, and text2image and compared in terms of the metrics that are majorly indicative of accuracy in literature.
2. Each concept in SageAttention2 is tested separately for average accuracy and worst accuracy, while keeping the other parameters of the attention frozen. These include but not limited to:
```
    a. Smoothing of Q and K, separately and together and comparing against other methods of preserving INT4 accuracy, like SmoothAttn, Hadamard transformation etc.
    b. Differing the INT4 quantization granularities among per-tokens, per-thread, per-block, and per-tensor.
    c. Differing the PV matmul calculation with FP8 E4M3, FP8 E5M3, FP16, and INT8 precisions while keeping the KQ matmul  INT4 methodology the same.
```
3. For performance, varying the different techniques of employing per-thread granularity in quantization of QK, employing the 2-level accumulation strategy for PV matmul, and smoothing of Q. Each of these techniques is ablated against the baseline of INT4 QK (K smoothed), PV FP8 matmul (with default FP32 accumulation.)

**Other Comments Or Suggestions:**

1. Would like to see more comparisons between SageAttention2-4b and SageAttention-8b and SageAttention, the visual results on HunyuanVideo and CogvideoX seems to indicate the weakness of INT4 range. Maybe related to loss of information to INT4 range since the smoothed Q distribution seems to indicate a normal curve, so there might be some unrecoverable accuracy due to INT4 range.

2. Could also include few benchmarks against speech related tasks, like audio generation, TTS and/or speech-to-text.

3. The end-to-end generation latency of SageAttention2-4b seems to be on par with SageAttention2-8b for 4090 and L20 GPUs. Wheras the TOPS tell a different story. When it comes to performance, I suggest relying on end-to-end latency in milliseconds instead of TOPS. Since it aligns more with real-world use cases.

4. More detailed ablation studies needed for the 2-level FP32 buffer accumulation in regards to accuracy, like FP8 E4M3 2-level accumulation vs baseline FP8 E4M3.

**Other Strengths And Weaknesses:**

This paper provides a bright way towards deploying low-latency transformer models. With great INT4/8 and FP8 attention mechanism, lot of real world real-time deployments of LLM could be possible as long as hardware supports such tensor cores. The SageAttention2 INT4 Q*K matmul algorithm seems highly intuitive, and given the per-thread granularity, it seems to be an optimal way to design quantization groups, which takes into account both performance and accuracy.

The paper restricts itself to only one technique to deal with quantization outliers, namely, smoothing. It would be great to see a comprehensive analysis of more methods that don't introduce any overhead, but that could be better suited to handle outliers for a reduced quantization error due to INT4 precision. The fact that INT8 version of SageAttention-2 matches or is on-par with the full-precision attention in accuracy, seems to indicate that there is room for improvement in the QK INT4 matmul front.

**Questions For Authors:**

1. Is the reason for not including the SageAttention2-4b in H100 and H20 benchmarks due to lack of INT4 tensor cores in Hopper GPUs used?
2. If the above is true, would it be possible to pack 2 INT4 values in INT8 for the above 2 gpu architectures?
3. Would learning for or calibrating for these per-thread quantization scaling factors lead to reduce the gap between SageAttention-8b and SageAttention-4b instead of calculating the scaled-max-of-absolute values as scaling factors?

**Relation To Broader Scientific Literature:**

More and more papers are targeting the GPU hardware by leveraging the instruction set, inherent knowledge of how GPU executes code blocks. We are seeing more of warp-level algorithms, algorithms designed to reduce any unnecessary communications overhead. Algorithm co-design with innate hardware knowledge are leading to better algorithms, that are turning out to be optimal.

This paper stands together with lot of papers that are structuring up the methodology of hardware-software co-design, which is a must for lot of real-world applications.

**Theoretical Claims:**

I checked for correctness of the following equations and their assumptions:
1. Q.K matmul, adjusting for smoothed Q and smoothed K. All conclusions are correct based on the formulations.
2. Online softmax equation adjusting for 2-level accumulation strategy with FP8 matmul
3. Optional smoothing of V, and adjusting for `V_mean` addition to the output.
4. Indices calculation for Q and K, along with the scaling factor calculations for the same, with correct assumptions for per-thread grouping of Q and K tokens.
5. The algorithm 1 in Page 4 seems to reflect the text astutely.

---

> ### Author Rebuttal · Authors · 2025-04-01
>
> Dear Reviewer 1zQu,
> Thank you for your valuable suggestions and questions. Below, we address each point raised.
>
> ---
> >### Comment1
>
> **Reply**: Thank you for your valuable suggestion. We compared the accuracy of sageattention, sage2-8b, sage2-4b, and sage2-4b without smooth Q across CogVideo layers (see table below). The limited INT4 representation range causes precision loss, but this is unrelated to Smooth Q. Before smoothing, the distribution was less uniform. While smoothing makes it closer to normal (though not perfectly uniform), it significantly reduces quantization error.
>
> |Attention|Cossim ↑|Relative L1 ↓|
> |-|-|-|
> | SageAttention|0.99995|0.00733|
> | Sage2-8b|0.99982|0.01573|
> | Sage2-4b|0.99460|0.06480|
> | Sage2-4b (without smooth Q)|0.9498|0.27305|
>
> ---
> >### Comment2
>
> **Reply**: Thank you for suggesting benchmarks on speech-related tasks. In response, we evaluated Qwen-2-Audio 7B, a speech-to-text model, on the ASR task using the Librispeech test split and measured its performance with the WER metric. The results are presented below:
>
> |Attention|test-clean ↓|test-other ↓|
> |-|-|-|
> |Full-Precision|1.74|4.01|
> |HadamardAttn|1.77|4.05|
> |SmoothAttn|1.76|4.01|
> |SageAttention|1.74|4.02|
> |Sage2-4b|1.73|3.99|
> |Sage2-8b|1.72|4.03|
>
> We can see that SageAttention2 consistently outperforms the baselines, highlighting its effectiveness in audio-related models and benchmarks.
>
> ---
> >### Comment3
>
> **Reply**: We agree and recognize the significance of end-to-end latency for real-world performance evaluation, which is why we report it in our results. However, we emphasize that the end-to-end speedup depends entirely on:
> 1) The attention speedup.
> 2) The proportion of total latency attributed to attention.
>
> For example:
> - If a model spends 50s on attention and 50s on other operations, a **2×** attention speedup reduces latency by **25s** (total: 75s).
> - A **2.5×** attention speedup yields a **30s** reduction (total: 70s).
>
> Moreover, quantifying the attention kernel’s FLOPs is essential, as it directly measures computational efficiency and our method’s advancement. This approach aligns with the evaluation standards of FlashAttention 1/2/3.
>
> ---
> >### Comment4
>
> **Reply**: Thank you for your valuable suggestion. We compare the accuracy of Sage2-8b with and without 2-level FP32 buffer accumulation across all layers of CogvideoX:
>
> |Model|Cossim ↑|Relative L1 ↓|
> |-|-|-|
> |Sage2-8b|0.9997|0.02133|
> |Sage2-8b (without two-level accumulation)|0.9939|0.17843|
>
> ---
> >### Question1
>
> **Reply**: Thank you for your question. Yes, the reason for not including SageAttention2-4B in H100 and H20 benchmarks is that Hopper GPUs do not have INT4 Tensor Core.
>
> ---
> >### Question2
>
> **Reply**: Thank you for your question. H20 and H100 are both Hopper architecture. Yes, it is technically feasible to pack INT4 into INT8, dequantize them to INT8 in the kernel, and use INT8 Tensor Core ops. However, this approach has significant drawbacks:
>
> 1. **Speed Impact**:
>    - It requires additional CUDA Core operations for dequantization and has the same Tensor Core count as INT8. Therefore it won't bring speed gains.
>
> 2. **Accuracy Impact**:
>    - INT4 quantization introduces greater accuracy loss than INT8.
>
> ---
> >### Question3
>
> **Reply**: Thank you for your valuable suggestion. We believe that learning the scaling factors would not help reduce the gap. Learned scaling factors are static and input-independent. However, the inputs exhibit significant fluctuations [1, 2], making static scaling suboptimal. For example, if all inputs are smaller than the predetermined scales, the representation range of INT4 is not fully utilized. Moreover, calibrating will compromise true plug-and-play compatibility.
>
> A more promising approach may be learning clipping factors [3]. It scales the absolute maximum by a learned ratio, which helps suppress outliers while preserving the representation range.
>
> ---
> [1] AWQ: Activation-aware Weight Quantization for LLM Compression and Acceleration
> [2] SmoothQuant: Accurate and Efficient Post-Training Quantization for Large Language Models
> [3] OmniQuant: Omnidirectionally Calibrated Quantization for Large Language Models
>
> ---
> ---
> If you feel your concerns have been resolved, we would greatly appreciate it if you consider raising the score.

---

> > ### Comment · Reviewer_1zQu · 2025-04-03
> >
> > ```
> > We agree and recognize the significance of end-to-end latency for real-world performance evaluation, which is why we report it in our results. However, we emphasize that the end-to-end speedup depends entirely on:
> >
> >     The attention speedup.
> >     The proportion of total latency attributed to attention.
> > ```
> > Hi, is it possible to update this analysis in the paper? The analysis on the breakdown of both INT4/8 QK preprocessing and attention kernels in terms of latency (in ms, not TOPS), along with similar analysis on the PV kernels would be really helpful to get a good understanding of the hotspots in end-to-end latency.
> >
> > Please also include the new tables from these rebuttals.

---

> > > ### Author Response · Authors · 2025-04-05
> > >
> > > Dear Reviewer,
> > >
> > > Thank you very much for your valuable suggestion! First, we provide a rough analysis of the overhead in milliseconds for a sequence length of 16K, as shown in the table below. We will analyze the INT4/8 QK and FP8 PV preprocessing and the attention kernel latency (measured in milliseconds) in detail in our paper.
> > >
> > >
> > > | Component             | Latency|
> > > |-----------------------|-------|
> > > | Original Attention        | 102.3 |
> > > | Sage2-8b | 39.5  |
> > > | Sage2-4b | 35.6  |
> > > | Smooth Q              | 3.9   |
> > > | Smooth K              | 0.49  |
> > > | Quant P               | 1.3   |
> > > | Quant V               | 2.5   |
> > > | INT4/INT8 Quant QK    | 2.7   |
> > >
> > > ---
> > > Furthermore, we will add the Tables from the rebuttals, as well as the new analyses and corresponding experiments, into our paper.
> > >
> > > ---
> > > We hope our reply can address your concerns. We would greatly appreciate it if you consider raising the score.

---

### Decision · Program_Chairs · 2025-05-01

**Decision:**

Accept (poster)

**Comment:**

Employing low precision MMA on new hardware for key operations in neural networks can improve model inference performance. Previously, SageAttention accelerates the attention operation by quantizing and using INT8 MMA for QK MatMul, and performing FP16 MMA (FP16 accumulation) for the PV MatMul, which is quite suitable for the platforms that supports INT8 and FP16 MMAs, such as NVIDIA Ampere GPUs. Later, new platforms that supports more advanced low precision MMAs, such as INT4 and FP8, have been released. For instance, NVIDIA Ada GPUs (SM89) support INT4, INT8, and FP8 MMAs, and NVIDIA Hopper GPUs (SM90) support INT8 and FP8 MMAs. Alternative attention acceleration techniques, such as FlashAttention, have been extended to support these new low precision MMAs and have achieved better performance. Therefore, it is worthwhile to explore the possibility of using even lower precision MMAs, including INT4 and FP8, for SageAttention to further improve the performance of the attention operation. This paper proposed SageAttention2, which is a continuation of the previous SageAttention work for new hardware platforms.

In this paper, the attention QKV data distribution has been studied to explore the way of performing quantization that balances the performance and accuracy. Building on top of the K quantization smoothing in SageAttention, tha paper found it is vital to do quantization smoothing for Q as well for SageAttention2 that performs compute in low precision.

Due to the QK MatMul for SageAttention2 is computed using INT4 MMA, the per-block quantization used for the INT8 QK MatMul in SageAttention would result in a large quantization error, whereas the much finer per-token quantization would result in lower performance due to each thread in a warp has to handle multiple scale factors in the implementation. The paper analyzed PTX MMA instruction layout requirements and found that by using the mma.m16n8k64 warp instruction and grouping the tokens from Q and K in a certain way, which is finer than per-block but coarser than per-token, each thread in a warp only has to handle one scale factor. This is equivalent as improving the QK quantized MatMul accuracy without sacrificing performance and is critical for the success of SageAttention2.

When it comes to the PV MatMul for SageAttention2, the paper found that using FP8 MMA results in similar accuracy as FP16 MMA, while the performance is significantly improved. An additional key finding from the paper is that the paper found the FP8 MMA(f32f8f8f32) instruction on the Ada and Hopper architecture is actually FP22, which can cause accuracy decay for PV compute. To mitigate this issue, the paper implemented a two-stage accumulation strategy, that is using FP22 (the apparent data type is still FP32 though) for the atom FP8 MMA accumulation (it has to be like this because of the MMA instruction), and use FP32 for the final accumulation in a separate buffer using accumulation instructions that have no such FP22 issue.

SageAttention2, developed specifically for the Ada architecture, surpass FlashAttention2 and xformers, originally developed for the Ampere architecture, by about 3x and 4.5x measured on RTX4090. On the Hopper architecture, despite the fact that INT4 MMA is not supported, SageAttention2 was adapted to use INT8 MMA for QK MatMul. The performance of SageAttention2 on the Hopper architecture is on par with FlashAttention3, which is originally developed for Hopper architecture, while delivering much higher accuracy.

Most of the questions from the reviewers are related to or originated from the following two aspects:

1. The motivation of some design choices, such as data types, quantization granularity, and quantization smoothing necessity, etc., which requires experimental result support in the presence and absence of those choices.
2. The lack of introduction of supported precisions on different hardware platforms. Many reviewers were not aware that the Hopper architecture, even though newer than Ada, does not support INT4 MMA, and they had been asking why some implementations such as SageAttention2-8b is necessary and why some experiments were missing for SageAttention2-4b on Hopper architecture, etc. The authors should clarify this in the introduction and related work sections.

The authors of the paper have adequately addressed the comments and concerns from the reviewers and will add the missing experimental results and clarifications in the final version of the paper as requested.

Overall, this is a solid paper that presents the improvement of a previous accelerated computing algorithm for a new hardware platform. It will have a somewhat large impact to the deployment of Transformer models and it should be accepted by ICML 2025.